# COMPLEXITY LAW:
# PAVING THE WAY FOR TIME SERIES FORECASTING

## ABSTRACT

Deep time series forecasting has emerged as a booming direction in recent years. Despite the exponential growth of community interests, researchers are sometimes confused about the direction of their efforts due to minor improvements on standard benchmarks. In this paper, we notice that, unlike image recognition, whose well-acknowledged and realizable goal is 100% accuracy, time series forecasting inherently faces a non-zero error lower bound due to its partially observable and uncertain nature. To pinpoint the research objective and release researchers from saturated tasks, this paper focuses on a fundamental question: how to estimate the performance upper bound of deep time series forecasting? Going beyond classical series-wise predictability metrics, e.g., ADF test, we realize that the forecasting performance is highly related to window-wise properties because of the sequence-to-sequence forecasting paradigm of deep time series models. In this paper, we delve into univariate time series forecasting, which is a prevalent forecasting paradigm spanning traditional statistical models to advanced time series foundation models. Based on rigorous statistical tests of over 4700 newly trained deep forecasters, we discover a significant exponential relationship between the minimum forecasting error of deep models and the complexity of window-wise series patterns, which is termed the *complexity law*. The proposed complexity law successfully guides us to identify saturated tasks from widely used benchmarks and derives an effective training strategy for large time series models, offering valuable insights for future research.

## 1 INTRODUCTION

In recent years, a significant number of deep learning models have been introduced for time series forecasting (Zhou et al., 2021; Wu et al., 2021; Wang et al., 2024d), which are equipped with elaborative architectures and meticulously crafted designs, demonstrating notable performance across diverse real-world forecasting scenarios, including finance (Gao et al., 2023; Wang et al., 2024c), transportation (Yu et al., 2017; Guo et al., 2019), and meteorology (Wu et al., 2023b; Wang et al., 2024b). Despite these advancements, we notice that the latest proposed models have shown minor improvements on existing widely used benchmarks. As presented in Figure 1, the improvement in the performance of deep time series models on four standard benchmarks has slowed significantly over the past three years. For instance, on the ETT benchmark (Zhou et al., 2021), the relative forecasting performance improvements exhibited a continuous downward trend from 2022 to 2025, with values of 14.98%, 7.77%, 3.93%, and 3.51% respectively. This stagnation has left the community increasingly confused about the research direction they are pursuing, as well as questioning the value of continuing to strive for minor improvements on these benchmarks.

Drawing inspiration from other tracks of the machine learning community like compute vision, we notice a fundamental difference between time series forecasting and these evergreen fields: *does the field have a well-known and clear performance upper bound*. For instance, image recognition in computer vision (Deng et al., 2009), a popular and long-standing area since 2000, has a quantifiable and widely accepted goal, that is, to achieve 100% accuracy. Besides, human performance in recognition can also serve as a good reference. Although the 100% accuracy may be practically unattainable due to labeling noise and inherent ambiguity in some images, these well-established targets not only provide a concrete reference of model capability but also ensure the community is clear on whether one task or benchmark has been well solved, fostering rapid progress and guiding

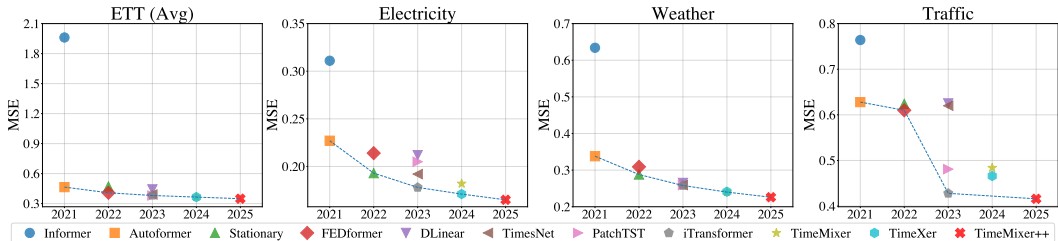

Figure 1: The performance change of deep time series forecasting in the past five years on well-established benchmarks. We record the MSE averaged from four widely used forecasting horizons.

researchers to focus on broader and harder tasks (Krizhevsky et al., 2012; Simonyan & Zisserman, 2014; He et al., 2016) when one task has been saturated. Unfortunately, there is no such clear goal for time series forecasting. Given the partially observable and inherently uncertain nature of time series, achieving zero prediction error is fundamentally unattainable. More importantly, even human experts often struggle to define what constitutes the best possible prediction for a given time series, making this area suffer from saturated tasks and measuring actual model quality.

Beyond general time series forecasting, some real-world applications exist with domain-specific "forecasting performance upper bound" stemming from years of expert experience, such as weather forecasting (Wu et al., 2023b) and financial quantification (Yang et al., 2020). However, such domain knowledge relies on human experts and is limited to specific areas, which cannot serve as a general rule for diverse and complex forecasting tasks focused on by the research community. Additionally, expert knowledge may result in a wrong judgment of the bottleneck due to personal bias or human limitations, which can be potentially broken by data-driven methodology (Bi et al., 2023). Motivated by the above observations, we believe that it is of immense importance to establish a *performance upper bound* for data-driven models to guide future research on deep time series forecasting.

Inspired by statistical estimation theory (Kay, 1993; Cramér, 1999), which rigorously defines the fundamental limits on the precision of any estimator, we argue that there indeed exists a general and instructive performance upper bound for time series forecasting, which is determined by the inherent predictability of time series. Through thousands of empirical experiments and rigorous statistical tests, we narrowed down a huge hypothesis space and finally discovered the *complexity law* of deep time series forecasting. Specifically, we observe a significant exponential relation between the intrinsic complexity in the temporal pattern of a given time series and its minimum forecasting error achieved by state-of-the-art deep models. Here, the temporal pattern complexity is defined in a window-wise format according to the sequence-to-sequence deep forecasting paradigm. This complexity law provides a principled framework for understanding the limits of time series predictability and sheds light on future research directions. Based on the newly proposed complexity law, we successfully identify the saturation of tasks within widely used benchmarks, releasing the research community from diminishing returns on over-studied datasets. Besides, the proposed law could seamlessly derive a simple yet effective training strategy by leveraging its quantitative results, which further boosts emerging large time series models. Our contributions are summarized:

- We notice a fundamental issue in the community of deep time series forecasting, that is, the absence of a general forecasting error lower bound for deep time series models.
- We find and formally define the *complexity law* for deep time series forecasting based on large-scale experiments. This law reveals that the minimum forecasting error achievable by deep models exhibits an exponential relationship with the window-wise pattern complexity of time series, which demonstrates favorable generalizability across diverse domains.
- Proposed complexity law provides valuable insights for the community, which can seamlessly support identifying saturated tasks and training deep and large time series models.

## 2 RELATED WORKS

### 2.1 TIME SERIES PREDICTABILITY

Time series predictability (Box et al., 2015) refers to the intrinsic degree to which future observations of a given series can be accurately inferred or estimated from past observations. It is a fundamental

concept in time series analysis that provides an a priori understanding of the best achievable complexity for a given time series, independent of the method. Consequently, accurately quantifying this property is a crucial prerequisite for evaluating the potential of the forecasting task and establishing a realistic performance benchmark. Over the past decades, researchers have developed various well-established statistical methods to quantify predictability (Pincus, 1991; Hamilton, 2020).

Traditional forecasting methodologies often rely on the assumption of stationarity (Granger IV et al., 2001). Therefore, assessing stationarity has become an important preliminary step in evaluating series predictability. The Augmented Dickey-Fuller (ADF) test (Elliott et al., 1992) is a statistical method for determining the time series stationarity. A smaller ADF-statistic, leading to the rejection of the null hypothesis, suggests a higher degree of stationarity, also potentially indicating higher predictability. Beyond stationarity, entropy-based measures offer a more concise way to quantify the complexity and predictability of a time series (Richman & Moorman, 2000). ForeCA (Goerg, 2013) introduces a quantitative measure of forecastability based on spectral entropy of the series. This metric can be calculated by subtracting the entropy of the series' Fourier domain, where a higher value signifies superior predictability, also offering a model-agnostic insight. However, most of the existing predictability measures are designed for an entire series, which is not consistent with the sequence-to-sequence forecasting paradigm for deep models. Further, the measurement of predictability in the context of deep time series forecasting remains relatively underexplored.

## 2.2 Time Series Forecasting Benchmarks

Evaluation benchmarks are fundamental in driving progress by providing researchers with clear and well-defined goals, which is acknowledged as one of the keys to "the second half of AI." With the rapid growth of time series forecasting models, a series of benchmarks has emerged to meet the growing need for rigorous and comprehensive model evaluations. For example, Informer (Zhou et al., 2021) introduced a standardized training and evaluation protocol for long-term time series forecasting, along with two widely used datasets, ETT and Electricity. Afterwards, Autoformer (Wu et al., 2021) introduced data from more diverse domains into this field, covering weather, health and finance. Besides, Solar-Energy (Lai et al., 2018) and PEMS for traffic are also evaluated (Liu et al., 2022; Wang et al., 2024b). These datasets span a broad spectrum of real-world applications and have been extensively adopted in subsequent research, promoting substantial development of the field.

Recently, with the advent of large time series models, fairly evaluating their capabilities, especially the zero-shot forecasting performance, has become a critical research focus. In response, several large-scale benchmarks have been developed. Woo et al. (2024) introduced the Large-Scale Open Time Series Archive (LOTSA), an extensive collection of billions of observations across nine domains. For probabilistic performance evaluation, Ansari et al. (2024) developed FEV, an open leaderboard dedicated to zero-shot forecasting. Aksu et al. (2024) proposed GIFT-Eval, a framework featuring distinct pretraining and train/test components tailored for evaluating pre-trained models.

While the above-mentioned benchmarks are widely used in current research, they only focus on the superficial data size or domain diversity during construction, lacking an in-depth analysis regarding the series' intrinsic characteristics, such as their inherent predictability. In this paper, we notice that only releasing the data and performance arena without a clear understanding of task predictability, research efforts risk being misdirected toward chasing marginal performance gains on simple or saturated benchmarks, thereby obscuring genuine advancements. Our complexity law can serve as a practical measurement to identify saturated tasks and provide insights to construct new benchmarks.

## 3 Complexity Law

As aforementioned, we attempt to build a general and instructive performance upper bound for deep time series forecasting, which is assumed to be relevant to time series predictability. To obtain a formal and statistically significant relationship, we start from a huge hypothesis space with a wide consideration of predictability and performance metrics, as well as relation types. However, since deep models typically employ a sequence-to-sequence forecasting paradigm, the classical series-wise predictability metrics fail in presenting a significant relation to final performance. Stemming from the deep forecasting paradigm, we newly propose a window-wise pattern complexity and finally establish an complexity law through thousands of experiments and rigorous statistical analysis.

## 3.1 HYPOTHESIS SPACE

Although the assumption of correspondence between time series predictability and the performance upper bound of deep forecasting seems reasonable, it is not easy to discover a general and significant law that can formalize the assumed correspondence. To narrow down the hypothesis space and isolate a significant relationship from intricate factors, our work is within the following scope:

*(i) Univariate forecasting.* In this paper, we only consider the univariate forecasting task. This scope can avoid the inherent performance trade-off in multivariate forecasting, allowing our statistical test to focus on temporal predictability. Besides, since most of the advanced deep time series models follow channel-independent training (Nie et al., 2022) and large time series models are all univariate (Shi et al., 2024; Liu et al., 2025), this scope will not damage the practicability of the conclusion.

*(ii) Approximated performance upper bounded.* Since the exact "optimal" forecast performance is unreachable, for each time series, we take the lowest error among several state-of-the-art forecasters as a surrogate. Despite this being only an approximation, the lowest value actually reflects the upper bound achieved by current research, which is already enough to identify which task falls beyond or behind the research frontier, as well as to reflect the task's inherent difficulty. More importantly, taking the lowest error among various models also offers a model-agnostic perspective for analysis.

Building up the above two basic settings, we examine a wide variety of performance and predictability metrics to characterize their underlying relationship (Fig. 2). The key factors can be organized in three dimensions:

**(i) Dim 1:** Predictability metric (*4 choices*): We consider three classical predictability indicators: Augmented Dickey-Fuller (ADF) test (Elliott et al., 1992) statistic, ForeCA (Goerg, 2013) and the half-life of the autocorrelation function (ACF) (Ramsey, 1974), which have served as the foundation of time series analysis. Additionally, we also proposed a window-wise complexity metric, which will be detailed later.

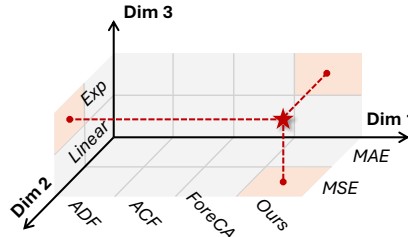

Figure 2: Hypothesis space considered in this paper. The discovered complexity law is marked with a red star.

**(ii) Dim 2:** Performance metric (*2 choices*): We adopt the most widely used metrics, Mean Absolute Error (MAE) and Mean Squared Error (MSE), to quantify the forecasting error.

**(iii) Dim 3:** Relation formalization (*2 choices*): Guided by the classical *principle of Occam's razor*, we only consider statistical tests on exponential[1] and linear relations, which are simple but sufficient.

After training 2,820 time series forecasting models as statistical samples and conducting rigorous statistical tests for the above 16 combined possible hypotheses, we have only observed one significant exponential relationship between our newly proposed complexity and MSE, which is named as *complexity law for deep time series forecasting*. This finding can be briefly described as follows.

> **Complexity Law (Informal).** *Within a certain interval of predictability, the lowest MSE of deep models exhibits a clear exponential relation with our proposed window-wise complexity.*

Next, we will present the definition of window-wise complexity and corresponding statistical results.

## 3.2 WINDOW-WISE PATTERN COMPLEXITY

According to our experiments that will be detailed in Section 4.1, all traditional series-wise predictability metrics (Elliott et al., 1992; Goerg, 2013) fail in demonstrating a value relationship. Thus, we attempt to present a new predictability metric tailored to the typical deep forecasting paradigm.

**Insights from forecasting paradigm** Typically, deep models utilize the sequence-to-sequence forecasting paradigm, where the model is expected to predict a future segment based on the latest segment of past observations (Zhou et al., 2021; Wu et al., 2021; Wang et al., 2024d). Specifically,

---

[1]The exponential relation test is conducted via the linearity w.r.t. Log Mean Squared Error (LogMSE) and Log Mean Absolute Error (LogMAE), which are defined as $\log(\text{MSE} + 1), \log(\text{MAE} + 1)$ to maintain the theoretical intuition that *the model performance metric should be zero when predictability metric is zero.*

given a time series $\mathbf{x} = \{\cdots, \mathbf{x}_t, \cdots\}$, its forecasting paradigms can be formalized as follow,

$$\text{Train: } \max_{\theta} \sum_t \mathbb{P}(\mathbf{x}_{t+1:t+F}|\mathbf{x}_{t-P:t}, f_\theta), \quad \text{Inference: } \widehat{\mathbf{x}}_{t+1:t+F} = f_\theta(\mathbf{x}_{t-P:t}), \quad (1)$$

where $f_\theta$ represents the forecasting model with parameter $\theta$. Here, $F$ and $P$ denote the lengths of the prediction horizon and past observation, respectively. The above paradigm motivates us that the predictability of forecasting models would be closely related to the window-wise property of time series rather than series-wise. Therefore, we present the following window-wise pattern complexity.

**Complexity definition**  Inspired by the above analysis, for each time series, we consider the continuous time windows with length $(P + F)$, which can completely cover all the time points used in a single forecast. Further, to highlight the variation pattern in time series (Wu et al., 2023a), we transform the time window into the frequency domain using the Fast Fourier Transform (FFT) and extract its amplitude spectrum only, which represents the magnitudes of all frequency components and demonstrates the strength of various variation patterns. This transformation also allows our metric to mitigate the influence of temporal shifts that are recorded in the phase information of the Fourier domain. To sum up, the computation process can be formalized as follows:

$$\{\mathbf{x}_{i:(i+P+F)}\}_i = \text{Split}(\mathbf{x}), \ \{\mathbf{A}_i\}_i = \{\text{Amp}(\text{FFT}(\mathbf{x}_{i:(i+P+F)}))\}_i, \quad (2)$$

where $\text{Amp}(\cdot)$ represents keeping the amplitude spectrum only and $\mathbf{A}_i \in \mathbb{R}^{P+F}$ denotes the extracted pattern of the $i$-th window. Based on this frequency domain representation, we define the series pattern complexity as the total variance of *amplitude spectrum distribution*, which records the spread of the amplitude spectra distribution among all time windows and inherently describes the distribution diversity. Thus, the proposed series pattern complexity of time series $\mathbf{x}$ is defined as:

$$\text{Complexity}(\mathbf{x}) = \text{tr}(\text{Cov}(\{\mathbf{A}_i\})) = \frac{1}{N} \sum_{1 \leq i \leq N} ||\mathbf{A}_i - \bar{\mathbf{A}}||_2^2. \quad (3)$$

Here, $\text{tr}(\cdot)$ denotes the matrix trace and $\text{Cov}(\cdot)$ denotes the covariance operator, where $N$ denotes the number of divided time windows and $\bar{\mathbf{A}}$ represents the sample mean of $\{\mathbf{A}_i\}$. The above complexity metric characterizes the intrinsic heterogeneity of series variations in every interested window, where a higher value suggests the series contains more distinct patterns and is harder to predict.

**Understanding window-wise pattern complexity**  We notice that the *transformation to Fourier domain* is key to the above definition, which also yields instructive insights for subsequent statistical tests. More empirical comparisons to prove its superiority can be found in the Section 4.1.

*(i) Joint distribution modeling.* As presented in Figure 3, we consider both past observations and the future horizon within the divided time windows. A simple option is to directly compute variance in the original time domain. However, since the time domain L2 distance is computed independently for different timestamps, the time domain variance can not capture the whole covariance matrix, thereby neglecting the mutual information between the past and future. In contrast, due to the *global nature of the frequency domain*, which we have carefully selected, our definition in Eq. 3 naturally captures the joint distribution of past and future. which can be interpreted as simultaneously characterizing both the initial state and the transition dynamics.

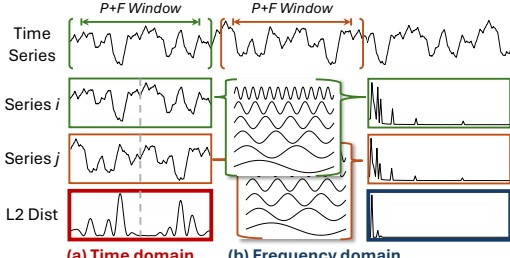

Figure 3: Illustration of time and frequency distance.

*(ii) Temporal shift invariant.* Benefiting from our design in Eq. 2, which only considers the amplitude spectrum in the frequency domain while ignoring the phase term, for any lag $\tau$, the series $\{\mathbf{x}_t\}$ and its $\tau$-lagged counterpart $\{\mathbf{x}_{t-\tau}\}$ exhibit the same complexity. This enables our metric to capture principal variation in time series and avoid noise, thereby better reflecting the inherent predictability.

### 3.3 STATISTICAL TEST

In our narrowed hypothesis space, we assume a fundamental exponential or linear relation between the predictability metric of a time series and its achievable optimal forecasting performance. To

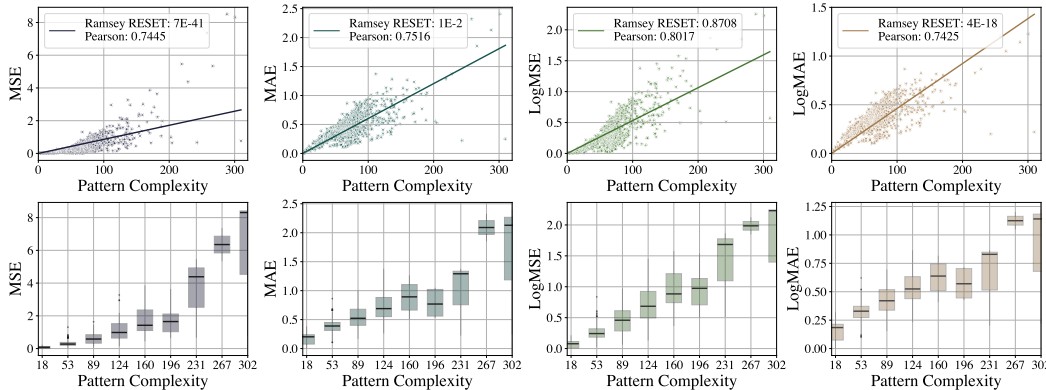

Figure 4: Experimental results from 940 series and 5 deep forecasters for the complexity law, which are processed from 4,700 experiments. For each data point, its $x$-coordinate denotes the window-wise pattern complexity calculated by Eq. 3 and the $y$-coordinate is the lowest forecasting error achieved among five experimental models: xPatch, TimeMixer++, TimeMixer, PatchTST, DLinear.

validate this, we conduct a systematic and comprehensive empirical evaluation of the relation between our newly defined window-wise pattern complexity (Eq. 3) and the minimum forecasting error achieved by several state-of-the-art deep forecasters. The following is the detail of our statistical test.

**Experimental data**    To ensure a comprehensive validation, we leverage the well-constructed public large-scale pre-training archive LOTSA (Woo et al., 2024) as our experimental subject. Concretely, LOTSA is a collection of time series datasets collected from real-world scenarios, spanning nine domains. To mitigate domain-level imbalance, we randomly sampled 20 time series from each dataset. Subsequently, we filter out series shorter than 5000 time steps to guarantee the training demand of deep learning models, resulting in a total of 940 univariate time series. Here, we treat each series as an independent task and train separate models on a series-by-series basis.

**Experimental models**    To measure the achievable forecasting performance, we include five state-of-the-art deep time series forecasters, including PatchTST (Nie et al., 2022), DLinear (Zeng et al., 2023), TimeMixer (Wang et al., 2024b), and TimeMixer++ (Wang et al., 2024a), and xPatch Stitsyuk & Choi (2025). These models have demonstrated exceptional performance on various benchmarks and are widely recognized for their effectiveness in univariate time series modeling.

**Implementation Details**    Following established conventions (Wang et al., 2024d), we split the entire time series into training, validation, and test sets with a ratio of 7:1:2 . Experimentally, we train and evaluate all three models on the selected 940 time series independently and record performance under the input-96-predict-96 setting. For each time series, we take the best performance achieved across five experimental models as one data point for subsequent statistical tests.

**Statistical analysis**    As shown in Figure 4, we identify a clear *linear relation between the proposed window-wise pattern complexity and the forecasting LogMSE*. To obtain a rigorous and formal description of the linear relationship, we formalize this observation from two perspectives.

*(i) Linear dependence test.* First, a linear regression analysis is conducted to quantify the dependence between pattern complexity and various evaluation metrics. Among them, LogMSE achieves the strongest linear association, with a Pearson correlation coefficient (Benesty et al., 2009) of 77.67%, which is generally considered to indicate a *strong positive linear correlation* in statistical domains.

*(ii) High-order dependence test.* Second, to avoid neglecting higher-order dependence, we further perform the Ramsey RESET test (Volkova & Pankina, 2013) on the second-order relation to examine whether higher-order terms are statistically required in precisely describing the relationship. Here, a lower p-value indicates greater confidence in high-order dependence. For LogMSE, the p-value is 0.85, well above the 0.05 significance threshold, indicating that introducing higher-order terms does not provide additional explanatory power. In contrast, the p-values for MAE, MSE and LogMAE are much smaller than 0.05, revealing the high-order dependence among these metrics and complexity.

**Summary**   Based on the above results, we successfully discovered a pure and confident exponential relationship between MSE and our proposed series complexity, formally defined in Eq. 4.

---

**Complexity Law.** *There exists an interval of window-wise pattern complexity* $(C_{\min}, C_{\max})$ *such that, for any time series* $\mathbf{x}$ *with* $\mathrm{Complexity}(\mathbf{x}) \in (C_{\min}, C_{\max})$, *the minimum forecasting MSE achieved by all feasible deep models admits a exponential relation with the complexity, which can be quantified as follows:*

$$\mathrm{MSE} \approx \exp\left(\alpha \cdot \mathrm{Complexity}(\mathbf{x})\right) - 1, \tag{4}$$

*where* $\mathrm{Complexity}(\mathbf{x})$ *represents our newly proposed pattern complexity measurement of time series* $\mathbf{x}$. *In our statistical test experiments above,* $C_{\min} = 0, C_{\max} = 309, \alpha = 0.0053$.

---

This complexity law provides an empirical framework for understanding time series predictability in the context of deep forecasting. Intuitively, time series with lower pattern complexity generally exhibit smaller forecasting errors, positioning them at the lower end of the fitted line and indicating better predictability. Importantly, the linear relation connecting these two regimes is non-trivial but has been rigorously validated through our statistical tests. Moreover, it is worth noting that the intercept of the complexity law equals zero, implying that as the complexity of a time series approaches zero, the infinity of its forecasting MSE under deep models also vanishes, which aligns with the intuition of perfect predictability. Further, the complexity law serves as both a predictive and prescriptive tool. By calculating the pattern complexity of time series, it becomes possible to estimate the expected forecasting error using this empirical relationship. This provides a crucial reference for evaluating model performance and offers actionable insights to guide future improvements.

**Remark 3.1** (The value of coefficient $\alpha$). *The complexity law in Eq. 4 is tested under the input-96-predict-96 scenario, which means changing the forecasting setting may affect the fitted relationship. Thus, we also experiment on different lookback and forecasting settings, which can be found in Appendix E. It can be observed that although the concrete value of $\alpha$ may vary, the experimental results demonstrate that the exponential relationship between MSE and pattern complexity is robust and general. It is really hard to identify a universal constant across various scenarios, and such a constant does not exist even in the scaling law of LLMs (Kaplan et al., 2020; Hoffmann et al., 2022).*

## 4   EXPERIMENTS

To further demonstrate the discovered complexity law's (Eq. 4) fundamentality and value to the deep forecasting research, this section will first make a comprehensive comparison with conventional predictability metrics and then showcase the insights for deep and large time series models.

### 4.1   OVERALL COMPARISON

As stated in Section 3.1, we widely consider a huge hypothesis space to discover the complexity law. Here, we present the statistical test on each hypothesis. Additionally, to examine the effectiveness of our proposed window-wise complexity, we also include the statistical test on some variants.

**Main results**   From the statistical results in Figure 5, we can obtain the following observations:

*(i) Comparison with conventional series-wise indicators.*   As aforementioned, the Augmented Dickey-Fuller (ADF) test (Elliott et al., 1992) statistic, and the ForeCA (Goerg, 2013) and the half-life of the autocorrelation function (ACF) (Ramsey, 1974) are considered in hypothesis space. Despite widespread adoption, we find that these series-wise predictability metrics do not produce a significant correlation with final forecasting performance in terms of the Pearson coefficient, as shown in Figure 5-left. This also indicates a long-standing gap between forecastability and forecasting methods, which can be partially closed by our proposed window-wise complexity.

*(ii) Ablations on proposed window-wise complexity.*   Here we also examine some variants of our complexity proposal, including changing the measure from the frequency domain to the time domain, such as window-wise Dynamic Time Warping (window DTW, (Berndt & Clifford, 1994)), as well as setting the time window size as past observation $P$ and forecasting horizon $F$. As illustrated in Figure 5-middle, the frequency domain measure significantly outperforms the time-domain DTW, highlighting the merits of our definition in series pattern recognition. While using the forecasting

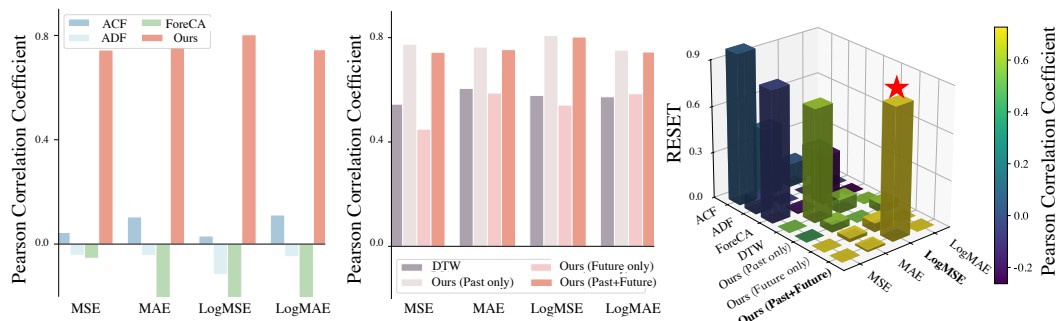

Figure 5: Comparison of our window-wise complexity with classical metrics (left) and other variations (middle), where Pearson coefficient between the predictability and performance is recorded. For the right part, the Ramsey RESET test is presented as the z-axis value; a higher bar indicates greater confidence in the linear relation; the brighter color refers to the higher Pearson coefficient.

window yields a higher Pearson coefficient, lower RESET test (Figure 5-right) results suggest the presence of high-order dependencies, indicating that the relationship cannot be considered linear. Besides, we can also find that our design in considering both observation and prediction window is better than solely considering one of them, since considering length-$(P + F)$ series allows a comprehensive modeling of both margin distribution and covariance information.

**Analysis of conventional series-wise indicators** As discussed before, the classical series-wise indicators do not match the sequence-to-sequence deep forecasting paradigm. Here, to enhance the classical indicators, we increase the input length of deep models to 512 and explore whether there is some correlation between ADF and performance. The result is in Figure 6, where still no obvious relation can be observed, highlighting the importance of window-wise complexity definition.

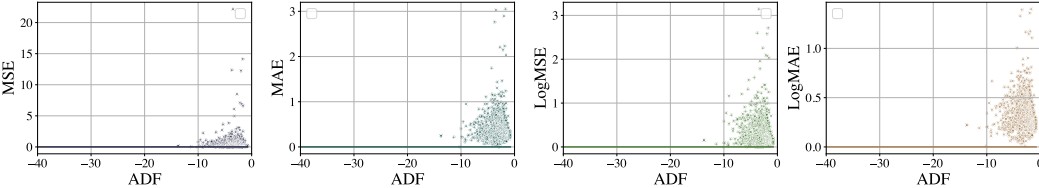

Figure 6: Relation between series-wise ADF and model performance under increased input length.

## 4.2 PRACTICE 1: IDENTIFY SATURATED FORECASTING TASKS

Forecasting benchmarks, such as ETT, Electricity and Weather (Zhou et al., 2021; Wu et al., 2021), have been essential for evaluating the performance of deep time series models. However, as presented in Figure 1, a notable trend has emerged that the performance gains of newly proposed models on existing long-term forecasting benchmarks have become increasingly marginal. These diminishing performance gains suggest these benchmarks may be approaching their predictability limits. Recognizing saturated tasks is crucial to enabling the community to shift focus from over-explored benchmarks and redirecting research toward more meaningful challenges. With the help of the complexity law, it is easy to obtain a clear vision for this fundamental question.

**Practical usage** Building on large-scale experiments, the complexity law offers a empirical framework to provide a estimation of forecasting error of a given time series, which can be used to identify saturated benchmarks. Specifically, by comparing the optimal performance of deep models on the interested benchmarks against this performance bound estimated by the complexity law, we can quantitatively assess whether a benchmark has reached its estimated intrinsic predictive limit. If the forecasting performance of the series falls far below the fitted complexity curve, it implies that further improvements are limited more by the data itself than by advances in model design.

**Results** Following the experiment protocol described in Section 3.3, we compute the pattern complexity and best forecast performance of each variable within well-established benchmarks. As illustrated in Figure 7, nearly all points in ETT, Electricity, Weather, and Exchange-Rate benchmarks

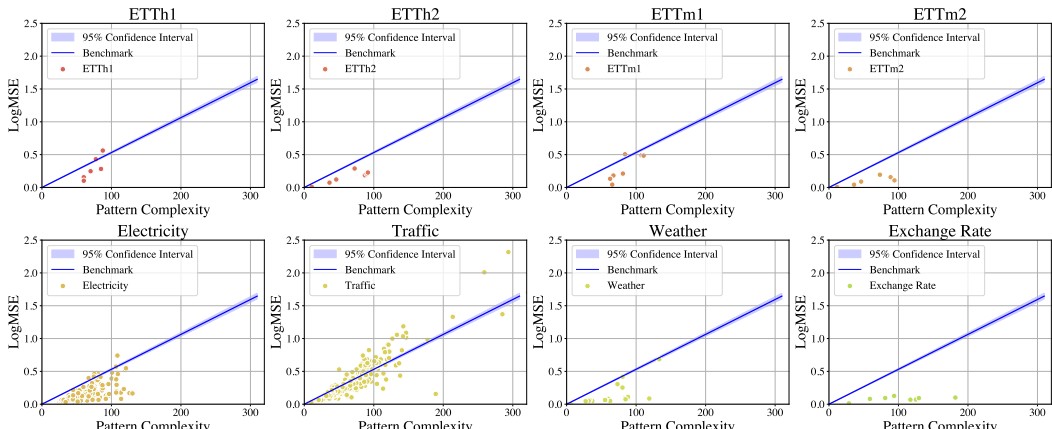

Figure 7: Analysis on widely used standard benchmarks. The scatter points represent individual time series from different benchmarks, and forecasting performance is measured by the minimum univariate forecasting error achieved by five tested state-of-the-art deep models as described in Figure 4. For a benchmark containing multiple series, if most series exhibit forecasting errors substantially below the estimation, we can consider this benchmark as saturated.

fall below the estimated performance bound, indicating that these datasets have likely reached their predictability limits. Besides, it is also observed that many variables in the Traffic benchmark are still above the estimated error, which may be because this task is highly spatiotemporal dependent. This finding also implies a promising direction in spatiotemporal forecasting.

> **Finding 1.** *According to our proposed complexity law, ETTh2, ETTm2, Electricity, Weather and Exchange-Rate can be considered as saturated, while Traffic still requires more investigation.*

### 4.3 PRACTICE 2: GUIDING LARGE TIME SERIES MODELS

For large time series models, despite extensive multi-domain data being adopted for pre-training, the existing pre-training corpora suffer from significant imbalances. For instance, in Time-MOE (Shi et al., 2024) (Table 1), nature data with moderate complexity dominate the pretraining corpus, whereas high-complexity subsets, such as web data, are severely underrepresented. This skewed distribution biases the models toward simpler temporal patterns, thereby limiting their ability to forecast complex dynamics. Moreover, existing zero-shot evaluation benchmarks (Aksu et al., 2024) often fail to expose this pretraining-complexity limitation. This is because both pretraining and evaluation datasets, typically sourced from real-world open data, are potentially in-domain, making it difficult to assess the generalization capability of large time series models in out-of-distribution (OOD) scenarios. This situation can be tackled through our exploration in a quantitative way.

Table 1: Window-wise complexity (Eq. 3) of Time-300B used in Time-MoE (Shi et al., 2024).

|  | Energy | Finance | Healthcare | Nature | Sales | Synthetic | Transport | Web | Other |
|---|---|---|---|---|---|---|---|---|---|
| Percent | 5.17 % | 0.0001% | 0.0001% | 90.50 % | 0.008 % | 2.98% | 0.69 % | 0.58 % | 0.006 % |
| Complexity | 162.465 | 149.815 | 138.781 | 150.945 | 127.710 | 179.309 | 114.705 | 173.587 | 158.634 |

**Usage 1: Constructing benchmark**  In pursuit of a more challenging OOD evaluation benchmark, finding a metric to quantify the pretraining-evaluation distribution gap is the beginning step. Surprisingly, we find that our proposed complexity metric (Eq. 3) can be naturally extended to measure the divergence between two time series, further facilitating the evaluation of the domain gap. Specifically, given two time series $\mathbf{x}, \mathbf{y}$, the extension of Eq. 3 can be defined as follows:

$$\text{Divergence}(\mathbf{x}, \mathbf{y}) = \frac{1}{NM} \sum_{i=1}^{N} \sum_{j=1}^{M} \|\mathbf{A}_i^{\mathbf{x}} - \mathbf{A}_j^{\mathbf{y}}\|_2^2, \text{ where } \{\mathbf{A}_i^*\}_i = \left\{ \text{Amp}\left(\text{FFT}\left(*_{i:(i+P+F)}\right)\right) \right\}_i.$$

Based on the above metric, we can quantitatively compute the divergence between the popular evaluation benchmark: GIFT-Eval (Aksu et al., 2024), and advanced pretraining datasets: Time-300B

(Shi et al., 2024) and TimeBench (Liu et al., 2025). Specifically, we randomly sample 100 series from each subdomain in the GIFT-Eval, Time-300B and TimeBench. Then, compute the pair-wise divergence between series in the pretraining and evaluation dataset. The histograms of computed divergence are plotted in Figure 8-left, corresponding to the blue part.

Adopting this measurement as a reference, to further eliminate potential domain overlap and enhance the generaliability test, we newly generated 100,000 sequences as a new evaluation benchmark following the generation rules proposed by Ansari et al. (2024). As demonstrated in Figure 8-left, compared to GIFT-Eval, our newly generated benchmark has much higher complexity divergence w.r.t. the pretraining dataset, which can provide a stricter evaluation of the generalization capability of large time series models, such as Time-MoE (Shi et al., 2024) and Sundial (Liu et al., 2025).

> **Finding 2.** *Current large time series model pretraining and evaluation are insufficient to test model generalizability. Generation data can enable a better leave-out evaluation.*

**Usage 2: Training strategy** Beyond constructing the evaluation benchmark, the complexity law can also be used to derive a simple yet effective sampling strategy to ease the imbalance between quantity and pattern complexity of pre-training data. Specifically, by assigning higher sampling weights to time series with greater complexity calculated by Eq. 3, the model is encouraged to learn more diverse and intricate temporal patterns during training.

To validate its effectiveness, we pre-train two representative large time series models, Time-MoE (Shi et al., 2024) and Sundial (Liu et al., 2025), which feature point-wise and patch-wise modeling architectures, respectively, on the TimeBench dataset. We then evaluate their performance using our newly proposed generative data benchmark. To ensure a fair comparison, we strictly adhered to their reported pretraining paradigms, modifying only the data sampling weights. As illustrated in Figure 8-right, our sampling strategy, derived from the complexity law, consistently enhances the performance of both models by a significant margin, demonstrating its generalizability and effectiveness.

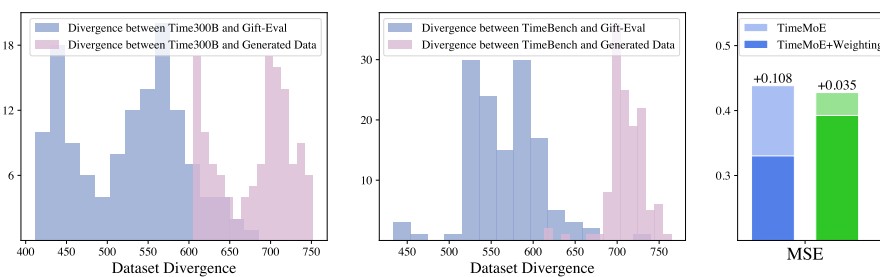

Figure 8: Practice of complexity law in large time series models. Left: Histogram of divergence between pre-training corpus (Time-300B and TimeBench) and GIFT-EVAL. Right: Performance comparison for weighted sampling strategy evaluated on our constructed generation data benchmark.

## 5 CONCLUSION AND FUTURE WORK

In this paper, we discover the complexity law of deep time series forecasting through thousands of experiments and rigorous statistical tests, which reveals a significant exponential relationship between the intrinsic window-wise complexity and the minimum achievable forecasting error of deep models. The proposed complexity law offers valuable insights for future research on deep time series forecasting, including identifying saturated tasks and facilitating large time series model training. To the best of our knowledge, this is the first work to empirically estimate the relation between series complexity and forecasting performance in the context of deep models.

Considering the experiment overload, we only consider a limited scope of hypothesis space, which still requires 4,700 experiments for the main result. In the future, we will extend the complexity law to more performance metrics and the multivariate or even multimodal forecasting scenarios. Besides, the same as scaling law of LLMs, findings in this paper are mainly based on empirical experiments; exploring the theoretical foundation of complexity law is also an important direction.

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

# A  SUPPLEMENT RESULTS FOR FIGURE 5

## A.1  EXPERIMENTAL RESULTS FOR CLASSICAL PREDICTABILITY METRICS

We visualize the experimental results from 940 series and 2,820 deep forecasters for three classical predictability metrics. As shown in Figure 9, the distribution of the points appears highly disordered, showing no clear correlation between these traditional metrics and the optimal forecasting error, even when considering more complex relations beyond the exponential and linear assumptions within our hypothesis space. This further validates the challenges and efforts of deriving the complexity law.

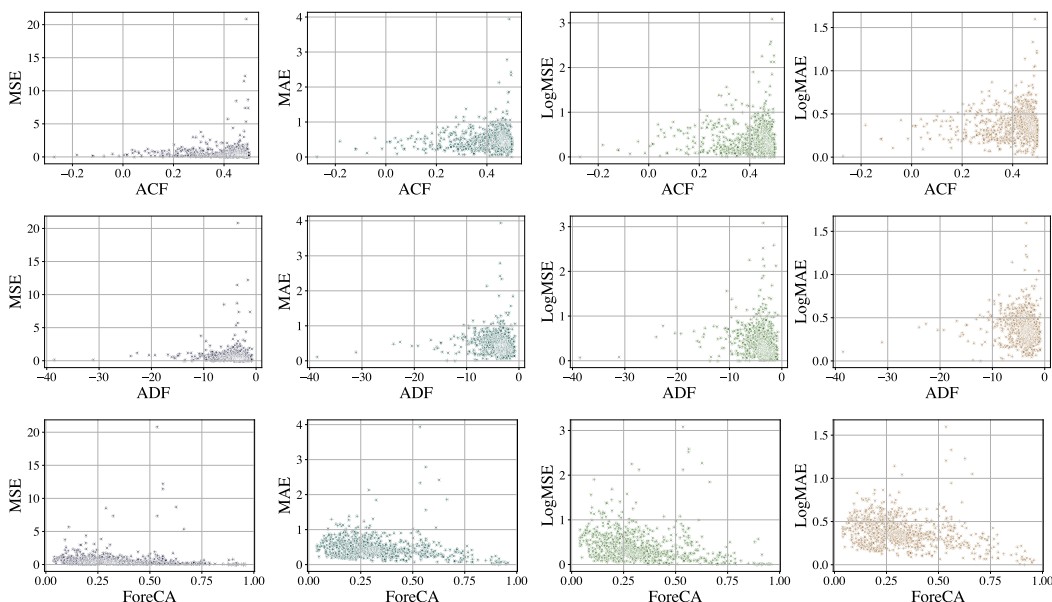

Figure 9: Experimental results between minimum forecasting error and classical forecasting metrics.

## A.2  QUANTITATIVE RESULTS

We perform extensive statistical analysis to examine the linear relationship between forecasting performance and different forecasting metrics, using the Pearson correlation coefficient (Benesty et al., 2009) and the Ramsey RESET test (Volkova & Pankina, 2013). The detailed results are presented in Table 2. A higher Pearson correlation coefficient indicates a stronger positive linear relationship. Beyond the linear dependence analysis, the Ramsey RESET test provides a more comprehensive evaluation of whether higher-order dependencies exist. Statistically, a p-value of 0.05 serves as the critical threshold for significance. When the p-value of the Ramsey RESET test is larger than 0.05, the underlying relationship can be described as a pure linear relation.

Table 2: Quantitative results of statistical test with different metrics. For clarity, we mark the value with  gray  if its Pearson Correlation coefficient is below 0.6 or the Ramsey RESET p-value falls below the 0.05 significance threshold, which means there does not exist a pure linear relation.

| Metric | Pearson Correlation | | | | Ramsey RESET | | | |
|---|---|---|---|---|---|---|---|---|
| | MSE | MAE | LogMSE | LogMAE | MSE | MAE | LogMSE | LogMAE |
| ACF | 0.0444 | 0.1040 | 0.0309 | 0.1115 | 9.78E-01 | 4.38E-01 | 1.53E-01 | 1.92E-01 |
| ADF | -0.0428 | -0.0436 | -0.1165 | -0.0487 | 9.23E-02 | 4.51E-04 | 9.28E-05 | 5.00E-05 |
| ForeCA | -0.0541 | -0.211 | -0.2283 | -0.292 | 8.53E-01 | 1.07E-02 | 3.17E-01 | 3.72E-04 |
| DTW | 0.545 | **0.6055** | 0.5786 | 0.5733 | 1.36E-31 | **7.34E-01** | 8.82E-02 | 2.03E-05 |
| Ours (Past Only) | 0.4492 | 0.5875 | 0.5416 | 0.5848 | 1.75E-19 | 4.95E-02 | 6.87E-05 | 5.50E-02 |
| Ours (Forecast Only) | 0.7751 | 0.7639 | 0.8079 | 0.7520 | 6.95E-54 | 2.20E-02 | 5.56E-02 | 3.34E-15 |
| Ours (Past+Forecast) | 0.7441 | 0.7540 | **0.8025** | 0.7450 | 2.16E-41 | 1.09E-02 | **8.52E-01** | 3.16E-18 |

# B SUPPLEMENT RESULTS FOR FIGURE 7

In this section, we present the detailed saturation analysis on existing long-term forecasting benchmarks. The results listed in Table 3 show that 100% of the variables in the ETTh2, ETTm2, Weather and Exchange-Rate benchmarks fall below the estimated performance bound, suggesting that these datasets can be considered saturated. In contrast, the saturation variables in the Traffic dataset remain below 80%, suggesting that this dataset still holds potential for further exploration and improvement.

Table 3: Saturation test results on existing long-term forecasting benchmarks, where we count the number of variables that fall below the performance estimated by the complexity law.

| Dataset | ETTh1 | ETTh2 | ETTm1 | ETTm2 | Electricity | Traffic | Weather | Exchange Rate |
|---|---|---|---|---|---|---|---|---|
| Total | 7 | 7 | 7 | 7 | 321 | 862 | 21 | 8 |
| Statured Count | 4 | 7 | 6 | 7 | 314 | 677 | 20 | 8 |
| Statured Ratio | 57.14% | 100% | 85.71% | 100% | 97.82% | 78.54% | 85.24% | 100% |

Additionally, we provide a box plot visualization comparing each benchmark with the data used in testing the complexity law, as shown in Figure 10. Notably, all variables in the Exchange-Rate dataset lie entirely below the lower quartile of the reference box plot distribution, underscoring its significant deviation from expected performance levels.

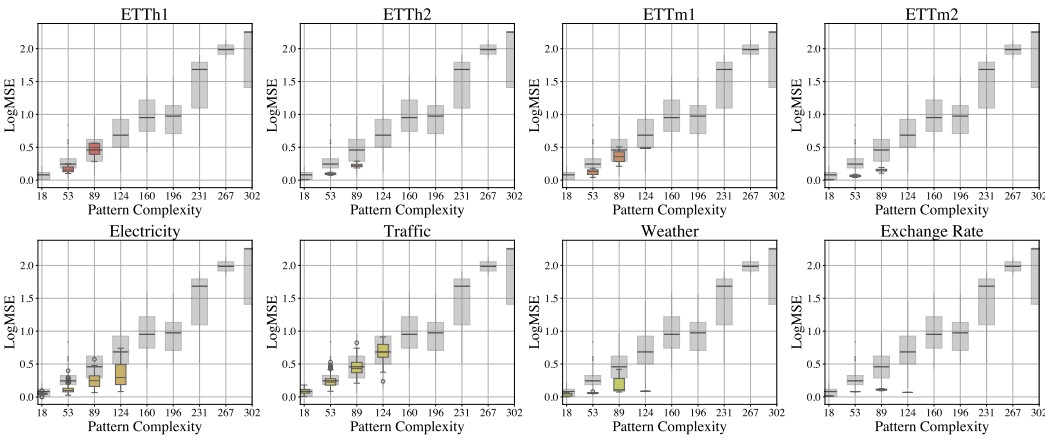

Figure 10: Saturation analysis under boxed visualization. Gray boxes refer to the data points from LOTSA (2024) used in computing the complexity law, while others refer to different benchmarks.

# C IMPLEMENTATION DETAILS

## C.1 CLASSICAL PREDICTABILITY METRICS

In our hypothesis space, we consider three conventional time series predictability metrics.

**ADF** The augmented Dickey–Fuller test (ADF) (Elliott et al., 1992) is a widely-used statistical test to assess the predictability of time series, as stationarity is a fundamental property of time series that significantly influences its predictability. For a time series $\mathbf{x}$, the ADF test operates under the null hypothesis that there is a unit root presents in time series using the following regression model:

$$\Delta \mathbf{x}_t = \alpha + \beta t + \gamma \mathbf{x}_{t-1} + \sum_{i=1}^{p} \phi_i \Delta \mathbf{x}_{t-i} + \epsilon_t \tag{5}$$

where $\alpha$ is a constant, $\beta$ is the coefficient on a time trend, and $\gamma$ is the coefficient testing the presence of a unit root. A significantly negative test statistic for $\gamma$ indicates rejection of the null hypothesis, suggesting stationarity. Time series deemed stationary by the ADF test are generally more predictable due to stable statistical properties over time.

**ACF**   The autocorrelation function (ACF) (Goerg, 2013) is a classical tool for measuring the degree of correlation between a time series and its lagged values over time, which is critical for determining its predictability. Given a time series $\mathbf{x}$, the autocorrelation with lag-$\tau$ is defined as:

$$\text{ACF}(\tau) = \frac{\text{Cov}(\mathbf{x}_t, \mathbf{x}_{t-\tau})}{\sqrt{\text{Var}(\mathbf{x}_t)\,\text{Var}(\mathbf{x}_{t-\tau})}}. \tag{6}$$

Here, we adopt the half-life length of the autocorrelation function defined above, which is the value of $\tau$, such that $\text{ACF}(\tau) = \frac{1}{2}\text{ACF}(1)$. This metric is widely used in financial analysis.

**ForeCA**   Given a time series $\mathbf{x}$, the forecastability (Goerg, 2013) can be quantified by analyzing the degree of orderliness in the frequency domain. Specifically, it is computed by subtracting the entropy of the normalized amplitudes of the series in the frequency domain as follows:

$$\mathbf{A} = \text{Amp}\left(\text{FFT}\left(\mathbf{x}\right)\right), \mathbf{p}_k = \frac{\mathbf{A}_k}{\sum_{k=1}^{L}(\mathbf{A}_k)}, \text{ForeCA}(\mathbf{x}) = 1 - \text{Entropy}(\mathbf{p}), \tag{7}$$

where $\text{Amp}(\cdot)$ represents the amplitude spectrum and $\mathbf{A}_k$ denotes the intensity of the frequency-$k$ periodic basis function. The forecastability metric ForeCA is derived by subtracting the entropy from 1, with a value closer to 1 indicating a highly predictable time series, and a value closer to 0 indicating a highly chaotic or random series that is less predictable.

## C.2   STATISTICAL TEST

This section presents details of the statistical tests employed.

**Pearson correlation**   The Pearson correlation coefficient (Benesty et al., 2009) is widely used to quantify the strength and direction of the *linear relationship* between two variables, $\mathbf{x}$ and $\mathbf{y}$. It is calculated as the covariance of the two variables divided by the product of their standard deviations:

$$\text{Pearson}(\mathbf{x}, \mathbf{y}) = \frac{\sum_{i=1}^{N}(\mathbf{x}_i - \bar{\mathbf{x}})(\mathbf{y}_i - \bar{\mathbf{y}})}{\sqrt{\sum_{i=1}^{N}(\mathbf{x}_i - \bar{\mathbf{x}})^2 \sum_{i=1}^{N}(\mathbf{y}_i - \bar{\mathbf{y}})^2}}, \tag{8}$$

where $\bar{\mathbf{x}} = \frac{1}{N}\sum_{i=1}^{N}\mathbf{x}_i$, $\bar{\mathbf{y}} = \frac{1}{N}\sum_{i=1}^{N}\mathbf{y}_i$ are the sample means. The coefficient ranges from -1 (perfect negative linear relationship) to +1 (perfect positive linear relationship).

**Ramsey RESET test**   The Ramsey RESET test (Volkova & Pankina, 2013) is frequently used to detect model specification errors, such as omitted higher-order terms in a regression model. In our statistical tests, we use it to test whether the second-order term is statistically required to precisely describe the relationship. Concretely, the procedure of Ramsey RESET for the following linear regression model is shown here,

$$y = \beta_0 + \beta_1 x_1 + \beta_2 x_2 + \cdots + \beta_k x_k + \epsilon.$$

*Step 1:* Fit the linear regression model and compute the predicted value $\hat{y}$ from the model. Then calculate the $R_{\text{old}}^2$ from the linear model.

*Step 2:* Perform a regression with an extended model that includes the high-order predicted values and calculate the new $R_{\text{new}}^2$.

$$y = \beta_0 + \beta_1 x_1 + \beta_2 x_2 + \cdots + \beta_k x_k + \gamma_1 \hat{y}^2 + \gamma_2 \hat{y}^3 + \cdots + \gamma_m \hat{y}^m + \nu.$$

*Step 3:* The Ramsey RESET test statistic can be calculated through,

$$\text{Ramsey RESET} = \frac{(R_{\text{new}}^2 - R_{\text{old}}^2)/m}{(1 - R_{\text{new}}^2)/(n - k - m - 1)}, \tag{9}$$

where $n$ denotes the sample size, $k$ refer to the dimension in the linear model, and $m$ is the number of higher-order terms added to the extended model.

Theorectically, the statistic Ramsey RESET follows an $F$-distribution with $m$ and $(n - k - m - 1)$ degrees of freedom. To obtain the p-value, we compare the observed statistic to the standard F-distribution. If the p-value is less than the significance level (e.g., 0.05), we reject the null hypothesis and conclude that the original model may suffer from specification errors, such as missing important nonlinear terms or interaction effects. Conversely, there is insufficient evidence to suggest specification errors in the original model based on this test.

## D  INDIVIDUAL PERFORMANCE OF FIVE EXPERIMENTAL MODELS

In the experiments, the forecasting performance for each series was defined as the minimum forecasting errors across several advanced deep forecasters. Here we further explore the relationship between series pattern complexity and the forecasting performance of each individual model. We can find that the relation between complexity and performance persists. The Pearson correlation coefficients for the individual models are 0.7979, 0.7772, 0.7647, 0.7236, and 0.7915 respectively.

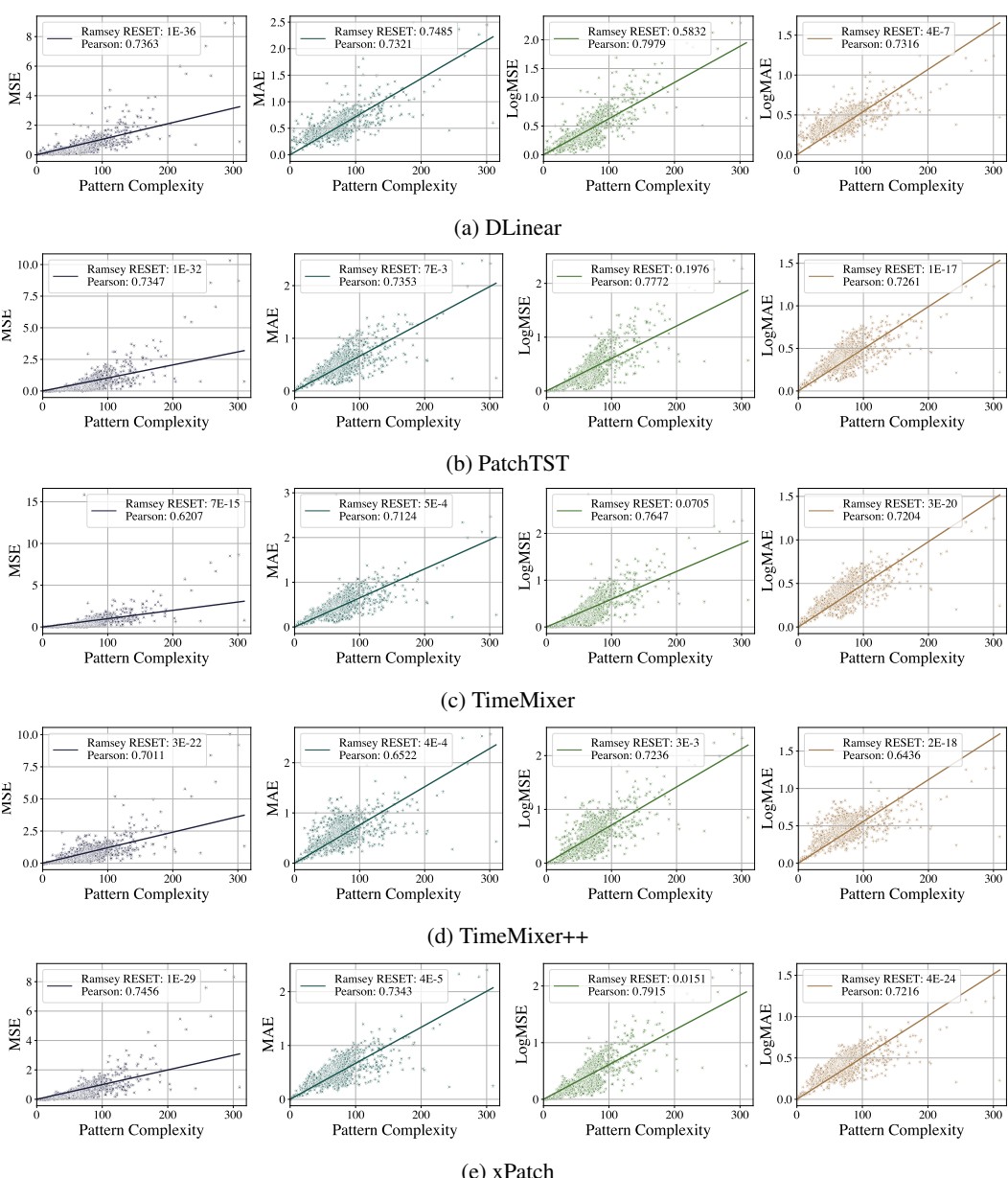

Figure 11: Forecasting performance and statistical test under the input-192-predict-96 setting based on 5 select state-of-the-art deep time series models, which is in the vertical layout.

## E  GENERALITY ANALYSIS

In the main text, we validated the proposed complexity under the input-96–predict-96 setup. To further validate the generality and robustness of our finding, extend the input length and the forecasting

horizon, respectively. Figures 12-13 present the experimental results and fitted curves across varying input and output lengths. Notably, the proposed complexity law, namely, the exponential relationship between series pattern complexity and forecasting performance, is clearly manifest. Notably, as shown in Figure 12, increasing the input length results in a decrease in the estimated coefficient $\alpha$. This aligns with the intuition cause longer input series typically yield better predictions.

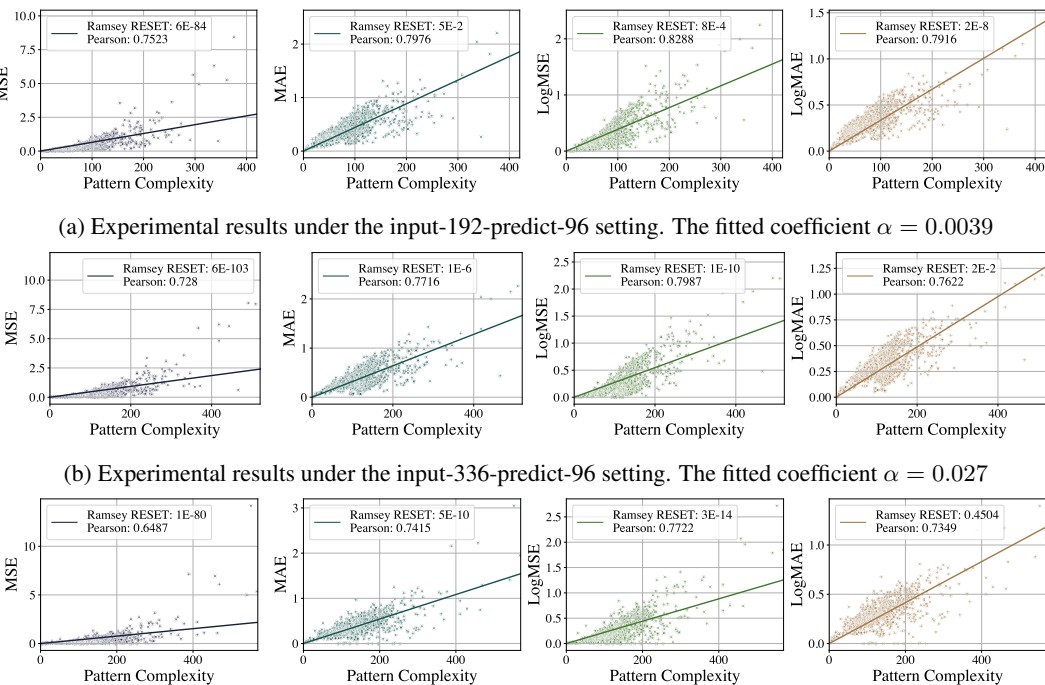

(a) Experimental results under the input-192-predict-96 setting. The fitted coefficient $\alpha = 0.0039$

(b) Experimental results under the input-336-predict-96 setting. The fitted coefficient $\alpha = 0.027$

(c) Experimental results under the input-512-predict-96 setting. The fitted coefficient $\alpha = 0.0022$

Figure 12: Fitted complexity law with increased look-back length in $\{192, 336, 512\}$.

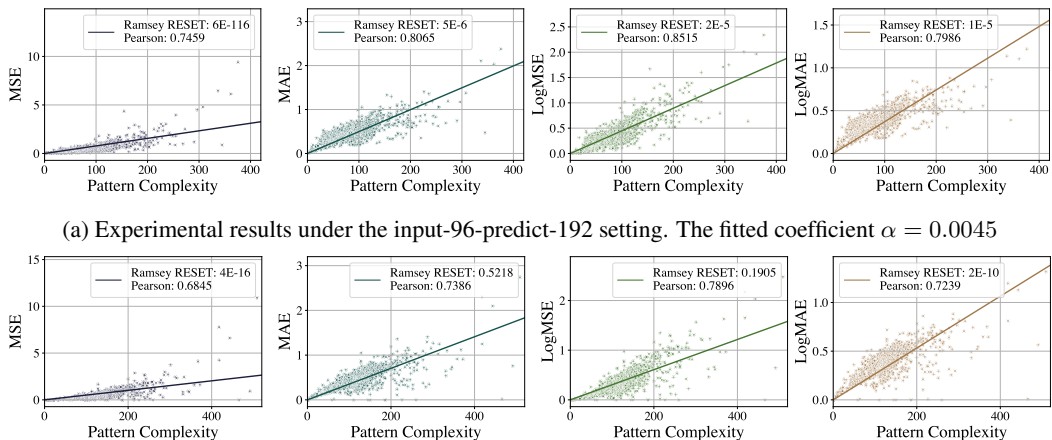

(a) Experimental results under the input-96-predict-192 setting. The fitted coefficient $\alpha = 0.0045$

(b) Experimental results under the input-96-predict-336 setting. The fitted coefficient $\alpha = 0.0030$

Figure 13: Fitted complexity law with increased forecasting length in $\{192, 336\}$.

# F  EXPERIMENTAL DATA ANALYSIS

## F.1  DATA SOURCE

As described in the main text, the 940 experimental time series is selected from LOTSA Woo et al. (2024). Here we list the concrete data source in Table 4.

Table 4: Source of our experimental data. Here we list datasets selected from LOTSA Woo et al. (2024), which should contain more than 20 time series and longer than 5000 time steps. Here "M, H, D" refers to minutely, hourly and daily sampling frequency respectively.

| Dataset | Domain | Frequency | # Time Series |
|---|---|---|---|
| Loop Seattle | Transport | 5M | 323 |
| Los-Loop | Transport | 5M | 207 |
| PEMS03 | Transport | 5M | 358 |
| PEMS04 | Transport | 5M | 307 |
| PEMS07 | Transport | 5M | 883 |
| PEMS08 | Transport | 5M | 170 |
| PEMS Bay | Transport | 5M | 325 |
| Q-Traffic | Transport | 15M | 45,148 |
| LargeST | Transport | 5M | 42,333 |
| Traffic Hourly | Transport | H | 862 |
| SHMetro | Transport | 15M | 288 |
| BDG-2 Panther | Energy | H | 105 |
| BDG-2 Fox | Energy | H | 135 |
| BDG-2 Rat | Energy | H | 280 |
| BDG-2 Bear | Energy | H | 91 |
| Buildings900K | Energy | H | 1,792,328 |
| BDG-2 Bull | Energy | H | 41 |
| BDG-2 Hog | Energy | H | 24 |
| KDD Cup 2018 | Energy | H | 270 |
| KDD Cup 2022 | Energy | 10M | 134 |
| Low Carbon London | Energy | H | 713 |
| London Smart Meters | Energy | 30M | 5,520 |
| Residential Load Power | Energy | M | 271 |
| Residential PV Powe | Energy | M | 233 |
| GEF12 | Energy | H | 20 |
| Wind Farms | Energy | M | 337 |
| CMIP6-2000 | Climate | 6H | 1,351,680 |
| CMIP6-2005 | Climate | 6H | 1,351,680 |
| CMIP6-2010 | Climate | 6H | 1,351,680 |
| ERA5-1997 | Climate | H | 245,760 |
| ERA5-1998 | Climate | H | 245,760 |
| ERA5-1999 | Climate | H | 245,760 |
| ERA5-2000 | Climate | H | 245,760 |
| Subseasonal | Climate | D | 862 |
| Subseasonal Precipitation | Climate | D | 862 |

## F.2  VISUALIZATION

Figure 14 illustrates the visualizations of time series data corresponding to varying levels of complexity. It is notable that as complexity decreases, the data becomes simple to learn. When the complexity reaches zero, the data reduces to a straight line, which aligns with intuitive expectations.

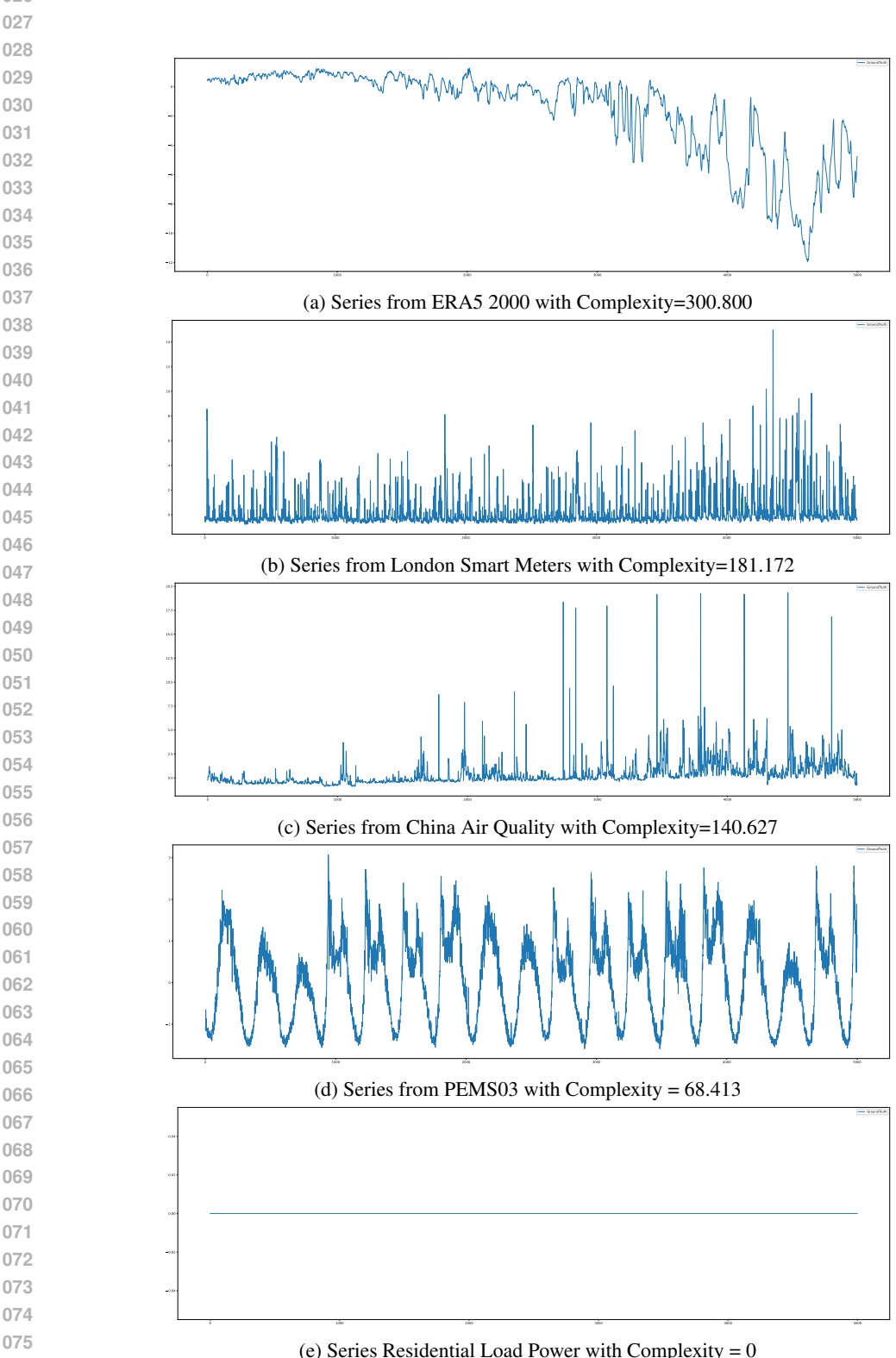

(a) Series from ERA5 2000 with Complexity=300.800

(b) Series from London Smart Meters with Complexity=181.172

(c) Series from China Air Quality with Complexity=140.627

(d) Series from PEMS03 with Complexity = 68.413

(e) Series Residential Load Power with Complexity = 0

Figure 14: Visualizations of time series examples with different pattern complexity values.

