# OpenReview forum: "Accuracy Law for the Future of Deep Time Series Forecasting"
_ICLR.cc/2026/Conference — Submitted to ICLR 2026_

### Official Review · Reviewer_DZKi · 2025-10-29

**Soundness:** 2
**Presentation:** 4
**Contribution:** 3
**Rating:** 6
**Confidence:** 4

**Summary:**

This paper reveals the Accuracy Law in the field of time series forecasting through extensive experiments, offering an empirically grounded and insightful principle with practical significance for the community.

**Strengths:**

The manuscript is well-written and effectively conveys its core ideas, enabling readers to clearly follow the authors’ reasoning. Moreover, the proposed predictability metric for time series is thoughtfully and ingeniously designed. Although the formulation is heuristic in nature, its validity is robustly supported by comprehensive experimental evidence presented in the paper.

**Weaknesses:**

**Clarity of Experimental Setup:**

The description of the experimental setup lacks sufficient detail in several key aspects:

1. The train/validation/test split ratios for the datasets are not disclosed, making it difficult to assess reproducibility and fairness of the evaluation.

2. It remains unclear whether the proposed *complexity* metric is computed on the test set alone or across the entire dataset. Regardless of the choice, a more thorough discussion of data partitioning is warranted. For instance, certain time series may exhibit high predictability in the training segment but low predictability in the validation or test segments—a scenario that could hinder model convergence and degrade test performance. The absence of such analysis weakens the practical utility and interpretability of the proposed *Accuracy Law*.

3. In Figure 5 (right), the term *Forecastability* is used without a formal definition or reference to its computation, which compromises clarity.

**Limited Generalizability of the Accuracy Law:**

1. While the authors present an impressive and commendable effort in formulating the **Accuracy Law**, its generalizability is constrained by the dependence of the complexity measure—and consequently the scaling exponent α—on the specific input–output window configuration. Given the prohibitive cost of conducting exhaustive experiments across all possible window settings, it is unreasonable to expect the research community to independently re-estimate α for every new forecasting horizon. At a minimum, the paper should provide pre-computed α values for commonly used input–output window pairs (e.g., 96→{96, 192, 336, 720} and 512→{96, 192, 336, 720}). Alternatively, the authors should propose a lightweight or analytical approximation method to estimate α efficiently, thereby enhancing the law’s practical applicability.

   In practice, recent time series models (e.g., xPatch, TimeMixer++) have achieved performance gains primarily on longer forecasting horizons such as 192, not on the 96→96 setting. Providing α only for 96→96 merely confirms a well-known fact, which—despite the paper’s good intent—limits its contribution.

2. For Section 4.3, Practice 2: Guiding Large Time Series Models, the base model’s prediction task is not inherently a 96-to-96 forecasting setup. When the input sequence is sufficiently long, restricting the input length to only 96 time steps becomes inappropriate, leading to two evident issues:

   1. Since the current accuracy law is exclusively defined for the 96-to-96 setting, its utility in guiding the construction of benchmarks (Usage 1) is severely limited. In the context of the present work, benchmark design should, at a minimum, not impose arbitrary constraints on input length—examples such as GIFT-Eval and the FEV Leaderboard underscore this need.

   2. Similarly, the applicability of this accuracy law to training strategy design (Usage 2) is also limited, as the actual training samples may not conform to a fixed 96-to-96 format. Moreover, the experimental setup described here lacks sufficient clarity; a detailed explanation—particularly regarding the choice of training sequence lengths—should be provided in the appendix, accompanied by further discussion on how varying input lengths impact model training.

**Questions:**

1. Please elaborate on the experimental setup and provide further discussion regarding dataset splitting and its complexity.

2. Include commonly used α values or a quick method for estimating α.

3. Provide further explanatory notes on the two usages in Practice 2.

Specific rationale is provided in the "Weaknesses" section.

I am willing to adjust my score based on the authors' response.

---

> ### Author Response · Authors · 2025-11-29
> **Response to Reviewer DZKi (Part 1)**
>
> Many thanks to Reviewer DZKi for providing a detailed review and insightful suggestions.
>
> > **W1:** **Clarity of Experimental Setup:**
> >
> > The description of the experimental setup lacks sufficient detail in several key aspects:
> >
> > Part 1 & Q1: The train/validation/test split ratios for the datasets are not disclosed, making it difficult to assess reproducibility and fairness of the evaluation. Please elaborate on the experimental setup and provide further discussion regarding dataset splitting and its complexity.
>
> Sorry for the missing details. For each series used in the experiments, we split the entire series into train/val/test with the ratio of 7/1/2 following the convention.
>
> > Part 2: It remains unclear whether the proposed *complexity* metric is computed on the test set alone or across the entire dataset. Regardless of the choice, a more thorough discussion of data partitioning is warranted. For instance, certain time series may exhibit high predictability in the training segment but low predictability in the validation or test segments—a scenario that could hinder model convergence and degrade test performance. The absence of such analysis weakens the practical utility and interpretability of the proposed *Accuracy Law*.
>
>
> Thank you for the insightful question. Since the forecasting MSE is evaluated on the test set, the complexity metric is also computed exclusively on the test set. Actually, the example you mentioned, where the training data exhibits high predictability while the test data shows low predictability, is indeed an important consideration. By computing complexity on the test set, we aim to capture the intrinsic predictability of the data specifically used for evaluation.
>
> > Part3: In Figure 5 (right), the term *Forecastability* is used without a formal definition or reference to its computation, which compromises clarity.
>
> Sorry for the misclarification. The Forecastability refers to ForeCA. We have revised this in the revision.
>
>
> > **W2 & Q2:** While the authors present an impressive and commendable effort in formulating the **Accuracy Law**, its generalizability is constrained by the dependence of the complexity measure—and consequently the scaling exponent α—on the specific input–output window configuration. Given the prohibitive cost of conducting exhaustive experiments across all possible window settings, it is unreasonable to expect the research community to independently re-estimate α for every new forecasting horizon. At a minimum, the paper should provide pre-computed α values for commonly used input–output window pairs (e.g., 96→{96, 192, 336, 720} and 512→{96, 192, 336, 720}).
> >
> > In practice, recent time series models (e.g., xPatch, TimeMixer++) have achieved performance gains primarily on longer forecasting horizons such as 192, not on the 96→96 setting. Providing α only for 96→96 merely confirms a well-known fact, which—despite the paper’s good intent—limits its contribution. Include commonly used α values or a quick method for estimating α.
>
> Following your suggestion, we newly included the two additional models (xPatch and TimeMixer++) into our experiments and conducted extensive experiments under varying input–output window setups, including increasing input length to 192, 336, 512, and increasing output length to 192 and 336, respectively. We have presented the results with the fitted $\alpha$ in $\underline{\text{Figures 12, 13 in the Appendix of the revised paper}}$. The results demonstrate that while the slope $\alpha$ does vary across different setups, the exponential relationship described remains consistent and robust.
>
> Also, we want to highlight that, even in the widely acknowledged scaling law, the constant of the fitted curve provided by DeepMind and OpenAI is different. Thus, identifying the exponential relation between MSE and pattern complexity is already a significant advance.
>
> [1] OpenAI, Scaling Laws for Neural Language Models, arXiv 2020
>
> [2] DeepMind, Training Compute-Optimal Large Language Models, arXiv 2022

---

> ### Author Response · Authors · 2025-11-29
> **Response to Reviewer DZKi (Part 2)**
>
> > **W3 & Q3:** For Section 4.3, Practice 2: Guiding Large Time Series Models, the base model’s prediction task is not inherently a 96-to-96 forecasting setup. When the input sequence is sufficiently long, restricting the input length to only 96 time steps becomes inappropriate, leading to two evident issues:
> >
> > 1. Since the current accuracy law is exclusively defined for the 96-to-96 setting, its utility in guiding the construction of benchmarks (Usage 1) is severely limited. In the context of the present work, benchmark design should, at a minimum, not impose arbitrary constraints on input length—examples such as GIFT-Eval and the FEV Leaderboard underscore this need.
> > 2. Similarly, the applicability of this accuracy law to training strategy design (Usage 2) is also limited, as the actual training samples may not conform to a fixed 96-to-96 format. Moreover, the experimental setup described here lacks sufficient clarity; a detailed explanation—particularly regarding the choice of training sequence lengths—should be provided in the appendix, accompanied by further discussion on how varying input lengths impact model training.
>
>
> Sorry for the missing details. Practice 2 centers on guiding large time-series models pre-training using the complexity-based weighting strategy, specifically as it applies to pre-training data. Experimentally, instead of using the 96-to-96 window, we followed the input lengths of pre-training data described in their original papers. In particular, TimeMoE uses an input length of 4096, and Sundial uses 2880, as described in their public papers and official implementations.
>
> We would like to clarify that the question of how varying input lengths impact model training for these large models is not the scope of our study. Therefore, to ensure comparability and alignment with existing literature, we follow the original input lengths and pre-training paradigm in the experiments.

---

### Official Review · Reviewer_VUQD · 2025-10-31

**Soundness:** 3
**Presentation:** 3
**Contribution:** 3
**Rating:** 6
**Confidence:** 4

**Summary:**

This paper, observing less and less improvement made by recent work on time series forecasting tasks, propose a question: "How sature is time series forecasting task? What's the upper bound?" To study this, authors conceptually rely on the "internal inherity" of time series, and define the accuracy law of time series forecasting, providing insights for the community about how and what direction we should move on.

**Strengths:**

1. The most significant pro I consider is the question proposed. Though recently there has been talks and papers on why the time series community has gone wrong, for example we have seen questions on foundation models that lack context[1], we have seen papers pointing out that hyperparameters are very sensitive for time series forecasting[2], etc. However, I do think this paper proposes an important question: the "upper bound" reachable for the time series forecasting datasets perhaps has been reached, and further modification of model architecture targeting these datasets might not be that meaningful. Especially they show, for example, ETT datasets, Electricity, Weather and Exchange-Rate has been saturated.
2. The proposed accuracy law, based on the "pattern complexity" of windows, makes sense on theory and shows reasonable forecasting correlation coefficient on benchmarks.

**Weaknesses:**

1. The authors mainly focus on univariate forecasting with only time series contexts, without other contexts like [1] mentioned.
2. I think the small progress in the recent 2-3 years made on time series forecasting benchmarks can already provide the insight that these benchmarks are not reliable. Of-course it would be better for someone like the authors to provide in-depth analysis and clearly show that these benchmarks have been saturated.

**Questions:**

1. Some previous work similarly splits the prediction loss into Bias and Variance; i.e. Bias means the "upper-bound" of forecasting accuracy or "lower-bound" of prediction loss, Variance is determined by noise, model learning paradigm, etc., they claim that bias decreases with longer input horizon[3]. What's your comment on this? Would you find out that longer input horizon could lead to higher prediction upper-bound?
2. Some claim that context is needed for time series forecasting. For example, consider these two cases: stock prediction and virus spread condition prediction. At some point they might look very similar, so a foundation model could give similar prediction, leading to non-accurate prediction[1]. However, in practice we use other contexts: e.g. we use alphas for stock prediction, and we use some other data and physics model for virus spreading prediction. What's your opinion on this?
3. Recently the community thinks that we have to use new benchmarks with more contexts and modalities other than time series alone for forecasting. Forecasting from time series alone is not reliable nor reasonable. Do you think this would be a good approach? What does the "accuracy law" for these cases might look like?

Ref:

[1] Fundamental limitations of foundational forecasting models: The need for multimodality and rigorous evaluation, NeurIPS 24 workshop talk.

[2] Position: There are no Champions in Long-Term Time Series Forecasting, Arxiv.

[3] Scaling Law for Time Series Forecasting, NeurIPS 24 poster.

---

> ### Author Response · Authors · 2025-11-29
> **Response to Reviewer VUQD**
>
> Many thanks to Reviewer VUQD for providing a detailed review and insightful suggestions.
>
> > **W1:** "The authors mainly focus on univariate forecasting with only time series contexts, without other contexts like [1] mentioned."
> >
> > [1] Fundamental limitations of foundational forecasting models: The need for multimodality and rigorous evaluation, NeurIPS 24 workshop talk.
>
> Thank you for the valuable question. In this paper, we follow the conventional definition of forecastability, which focuses solely on the target time series data, to ensure alignment with prior statistical measurements. We agree that incorporating multimodal information, especially external textual data, could facilitate forecasting performance. However, the scarcity of large-scale multimodal datasets makes it challenging to conduct comprehensive experiments in this direction. We see this as an important avenue for future research.
>
>
> > **W2:** "I think the small progress in the recent 2-3 years made on time series forecasting benchmarks can already provide the insight that these benchmarks are not reliable. Of-course it would be better for someone like the authors to provide in-depth analysis and clearly show that these benchmarks have been saturated."
>
> Thank you for recognizing our interest. The noticeable stagnation of current forecasting benchmarks is the motivation for our work. Building on this observation, we find and formally define the accuracy law for deep time series forecasting based on large-scale experiments. Notably, in addition to the discovery, the proposed accuracy law successfully guides us to identify saturated tasks from widely used benchmarks and derives an effective training strategy for large time series models, offering valuable insights for future research.
>
> > **Q1:** "Some previous work similarly splits the prediction loss into Bias and Variance; i.e. Bias means the "upper-bound" of forecasting accuracy or "lower-bound" of prediction loss, Variance is determined by noise, model learning paradigm, etc., they claim that bias decreases with longer input horizon[3]. What's your comment on this? Would you find out that longer input horizon could lead to higher prediction upper-bound?"
> >
> > [3] Scaling Law for Time Series Forecasting, NeurIPS 24 poster.
>
>
> We would like to clarify that our study centers on the final test performance of trained models, which is often considered as test loss, which differs from the train loss studied in the scaling law. As per request, we increased the input horizon to 192, 336, 512, respectively, and the experimental results are presented in $\underline{\text{Figure 12 in the Appendix of the revised paper}}$. We find that the fitted exponent $\alpha$ becomes smaller for longer inputs, corresponding to lower test loss or better performance. This is reasonable and expected, since a longer historical context typically results in better forecasting performance.
>
> > **Q2:** "Consider these two cases: stock prediction and virus spread condition prediction. At some point they might look very similar, so a foundation model could give similar prediction, leading to non-accurate prediction[1]. However, in practice we use other contexts: e.g. we use alphas for stock prediction, and we use some other data and physics model for virus spreading prediction. What's your opinion on this?"
>
> Thank you for the interesting question. Most of the existing large time series models perform univariate forecasting, where their predictions are driven primarily by the observed series. Consequently, when two series exhibit similar temporal variations, such models will tend to generate similar forecasts.
>
> > **Q3:** "Recently the community thinks that we have to use new benchmarks with more contexts and modalities other than time series alone for forecasting. Forecasting from time series alone is not reliable nor reasonable. Do you think this would be a good approach? What does the "accuracy law" for these cases might look like?"
>
> We agree that incorporating external covariates typically improves forecasting performance. However, the number of covariates and the causal relevance of such auxiliary data vary substantially across domains. Accordingly, this work does not attempt to resolve all possibilities of predictability in time series forecasting. Following the conventional measurements of forecastability, which is computed solely based on a single series like ADF, we focus on univariate forecasting in this paper. In the context of multimodal forecasting, the "accuracy law" might need to include the causality analysis between different series.

---

### Official Review · Reviewer_B8Rp · 2025-10-31

**Soundness:** 2
**Presentation:** 3
**Contribution:** 2
**Rating:** 4
**Confidence:** 4

**Summary:**

This paper notices that recent studies have achieved minor improvements on standard benchmark datasets. To pinpoint the research objective and release researchers from these saturated tasks, this paper considers it essential to estimate the performance upper bound of deep time series forecasting. Based on a series of experiments, the paper empirically finds and formally defines an accuracy law for deep time series forecasting. This law reveals that the minimum forecasting error (MSE) achievable by deep models exhibits an exponential relationship with the window-wise pattern complexity of the time series. By leveraging the proposed accuracy law, this paper identifies saturated tasks in widely used benchmark datasets and derives an effective training strategy to improve the generalization capability of large-scale time series models.

**Strengths:**

1. This paper newly defines window-wise pattern complexity and series pattern complexity.
2. The figures are visually appealing, the overall organization is reasonable, and the references are appropriate.
3. This paper has good writing.

**Weaknesses:**

Please see the Questions.

**Questions:**

1. The proposed Accuracy Law in Eq. 4 and the key Figure 4 are solely derived from experiments on three models. I believe that both the experimental and theoretical evidence are insufficient:
(1) The proposed accuracy law is essentially a linear association fitted based on the currently limited experimental results, relying heavily on the performance of the selected models. In fact, we have not yet obtained a true forecasting performance upper bound that is relevant to time series predictability. If additional models and datasets are included in the statistical analysis, the key parameter α in Eq. 4 would change, and consequently, the accuracy law line would also shift. Therefore, I argue that relying on the proposed accuracy law cannot effectively identify saturated forecasting tasks. For instance, if the models used to fit the accuracy law line perform poorly, many models may fall below the line, but this does not necessarily indicate that the forecasting task for that specific series pattern complexity has reached saturation.
(2) According to the proposed accuracy law, “the minimum forecasting MSE achieved by all feasible deep models admits an exponential relation with the complexity.” This implies that time series with the same pattern complexity should exhibit similar or nearly identical lowest forecasting errors. However, as shown in Figure 4, even for univariate time series with the same pattern complexity, their lowest forecasting errors vary substantially. Moreover, as pattern complexity increases, forecasting errors become more scattered. This observation conflicts with the claimed accuracy law, and the paper does not provide any explanation for this phenomenon.
(3) The current experiments use only three models, which are not fully representative. Selecting different types of time series models may yield different results. For example, models can be based on different frameworks such as MLP-based or Transformer-based, and consequently, some model types may be more sensitive to dataset characteristics, the length of the prediction horizon and past observation. To conduct a systematic and comprehensive empirical evaluation, I suggest adopting a broader range of models and conducting more in-depth analysis.
(4) The calculation of the proposed series complexity is not clearly explained. Are the divided windows overlapping or non-overlapping?
What is the stride used? The current demonstrations and conclusions are based on the input-96-predict-96 setting. How do different partitioning schemes or window sizes affect the computed series pattern complexity?
2. What is the theoretical basis for the complexity definition? Why use the Fast Fourier Transform (FFT) instead of other methods?
3. Others:
(1) The paper states C_min=0, how many such special time series exist, and what are their characteristics?
(2）The paper states C_max=309, but Figure 4 shows that the maximum value on the x-axis is 302.
(3) Is there any model that performs relatively worse than others on low pattern complexity series but performs better on high pattern complexity series?

---

> ### Author Response · Authors · 2025-11-29
> **Response to Reviewer B8Rp (Part 1)**
>
> Many thanks to Reviewer B8Rp for providing the insightful review and questions.
>
> > **W1-part 1:** "The proposed accuracy law is essentially a linear association fitted based on the currently limited experimental results, relying heavily on the performance of the selected models. In fact, we have not yet obtained a true forecasting performance upper bound that is relevant to time series predictability. If additional models and datasets are included in the statistical analysis, the key parameter α in Eq. 4 would change, and consequently, the accuracy law line would also shift. Therefore, I argue that relying on the proposed accuracy law cannot effectively identify saturated forecasting tasks."
>
> Thank you for the insightful comment. The models we used for experiments are not arbitrary; they are well‑acknowledged state‑of‑the‑art forecasting models, which are consistently adopted as competitive baselines in recent works. Following the reviewer’s suggestion, we have expanded the model set by adding two additional baselines (TimeMixerPP and Model xPatch). As presented in $\underline{\text{Figure 4 of the revised paper}}$, the exponential relation persists, and the $\alpha=0.0053$ is nearly identical to $\alpha=0.0054$ in the original submission.
>
> We would like to highlight that our main contribution lies in **discovering an exponential relationship between forecasting performance and complexity, not any single $\alpha$ value.** As with other empirical laws (e.g., scaling laws [1,2]), fitted coefficients depend on the experimental model and data but do not negate the discovered functional relationship. We also clarify that forecasting performance below the fitted line is not automatically evidence of task saturation; the law should be seen as a representative empirical frontier that can be updated if substantially stronger models emerge. If a model’s performance lies above this frontier, it indicates the model has not yet fully exploited the predictable component in the data, and further improvements are likely possible. Conversely, if performance lies below the frontier, this provides evidence that the task may have reached an empirical predictive upper bound.
>
> [1] OpenAI, Scaling Laws for Neural Language Models, arXiv 2020
>
> [2] DeepMind, Training Compute-Optimal Large Language Models, arXiv 2022
>
>
> > **W1-part 2:** "According to the proposed accuracy law, “the minimum forecasting MSE achieved by all feasible deep models admits an exponential relation with the complexity.” This implies that time series with the same pattern complexity should exhibit similar or nearly identical lowest forecasting errors. However, as shown in Figure 4, even for univariate time series with the same pattern complexity, their lowest forecasting errors vary substantially. Moreover, as pattern complexity increases, forecasting errors become more scattered. This observation conflicts with the claimed accuracy law."
>
> Thank you for the insightful question.
>
> We would like to clarify that the proposed accuracy law or complexity law describes a statistical relationship between pattern complexity and forecasting performance, and therefore provides an estimation of forecasting error achievable by feasible deep models for a given complexity level. It does not claim that all time series with the same pattern complexity will reach the same error.
>
> Regarding the deviations observed in Figure 4, we believe they may arise from several factors. Firstly, although the test data share the same complexity, the corresponding training sets may not necessarily exhibit the same complexity due to potential underlying temporal shifts or distributional differences. This discrepancy could introduce variability in forecasting performance. Moreover, it is important to note that as the complexity increases, the space of possible patterns expands, resulting in greater heterogeneity in the characteristics of time series data. Consequently, the empirical performance variance broadens.

---

> ### Author Response · Authors · 2025-11-29
> **Response to Reviewer B8Rp (Part 2)**
>
> > **W1-part 3:** "The current experiments use only three models, which are not fully representative. Selecting different types of time series models may yield different results. For example, models can be based on different frameworks such as MLP-based or Transformer-based, and consequently, some model types may be more sensitive to dataset characteristics, the length of the prediction horizon and past observation. To conduct a systematic and comprehensive empirical evaluation, I suggest adopting a broader range of models and conducting more in-depth analysis."
>
>
> Thank you for the valuable suggestion. We would like to highlight that our original experiments already include a representative MLP-based model, DLinear, and a Transformer-based model, PatchTST. Following your suggestion, we have additionally included two more competitive baselines and expanded the experimental scope by enlarging both the prediction horizon and the look‑back length. We have updated $\underline{\text{Figure 4}}$. The fitted accuracy law has a slope of $\alpha = 0.0053$, nearly identical to the $\alpha$ = 0.0054 reported in the original submission, which supports the representativeness of our original model choices. Experimental results with increased input and output lengths are provided in $\underline{\text{Figures 12, 13 in the Appendix of the revised paper}}$, demonstrating that the observed relationship persists across diverse settings.
>
>
> > **W1-part 4:** "The calculation of the proposed series complexity is not clearly explained. Are the divided windows overlapping or non-overlapping? What is the stride used? The current demonstrations and conclusions are based on the input-96-predict-96 setting. How do different partitioning schemes or window sizes affect the computed series pattern complexity?"
>
>
> Sorry for the missing details, the data process follows the convention in previous works[1,2]: windows are overlapping with a stride of 1. To further clarify how window size affects the calculated complexity, we increased the input length from 96 to 192, 336, and 512 and recomputed the complexity values. We reported the key quantiles below.
>
> | Setup    | Mean       | Std        | 10%        | 90%        |
> |:--------:|:----------:|:----------:|:----------:|:----------:|
> | 96-96    | 74.261053  | 49.325269  | 31.445645  | 115.368846  |
> | 192-96   | 94.407854  | 72.266029  | 34.608010  | 160.755263  |
> | 336-96   | 133.588735 | 90.173485  | 57.577470  | 211.627248  |
> | 512-96   | 145.466566 | 99.658678  | 68.197810  | 225.149103  |
>
> [1] Zhou, Haoyi, et al. "Informer: Beyond efficient transformer for long sequence time-series forecasting." AAAI 2021.
> [2] Wu, Haixu, et al. "Timesnet: Temporal 2d-variation modeling for general time series analysis." ICLR 2023.
>
>
> > **W2:** "What is the theoretical basis for the complexity definition? Why use the Fast Fourier Transform (FFT) instead of other methods?"
>
>
> Thank you for the valuable comment. The proposed pattern complexity is an empirical metric rather than a theoretically derived one.  We have provided a discussion on the use of FFT in $\underline{\text{Line 221 of the original paper}}$ and further conducted an ablation study on the complexity measurement in $\underline{\text{Figure 5 of the original paper}}$, where the proposed complexity surpasses other measurements.
>
> We would like to highlight that identifying a statistically significant and interpretable complexity metric from a huge hypothesis space is a non-trivial achievement. However, we remain open to further improvement. We appreciate that if the reviewer has suggestions for alternative definitions or methods, we are eager to explore them in future work.
>
> > **W3-part 1:** "The paper states C_min=0, how many such special time series exist, and what are their characteristics?"
>
> Thank you for the insightful question. We indeed observe 8 cases with computed complexity of C_min = 0. We have provided visualizations of the series in the $\underline{\text{Figure 14 in the Appendix of the revised paper}}$. Theoretically, a complexity value of zero typically corresponds to a perfectly linear trend or a strictly periodic waveform without distortion.
>
> > **W3-part 2:** "The paper states C_max=309, but Figure 4 shows that the maximum value on the x-axis is 302."
>
> Thank you for your careful reading. We believe there might be a misunderstanding regarding Figure 4. Figure 4 (lower)  presents a boxplot summarizing the distribution of LogMSE within binned complexity intervals, where the x-axis represents the mean complexity value for each bin. While the maximum complexity value in the dataset is C_max = 309, the rightmost bin in the boxplot corresponds to a mean complexity of 302. We hope this clarifies the discrepancy.

---

> ### Author Response · Authors · 2025-11-29
> **Response to Reviewer B8Rp (Part 3)**
>
> > **W3-part 3:** "Is there any model that performs relatively worse than others on low pattern complexity series but performs better on high pattern complexity series?"
>
>
> Thank you for the interesting question. To investigate this, we visualized the relationship between pattern complexity and model performance for each model individually. These visualizations are included in $\underline{\text{Figure 11 of the Appendix of the revised paper}}$, where we did not observe any model that performs worse on low-complexity series but better on high-complexity series. Instead, the general trend holds consistently across all models, that is, forecasting MSE increases as pattern complexity increases, as also reflected in Figure 4.

---

### Official Review · Reviewer_cwRd · 2025-11-02

**Soundness:** 3
**Presentation:** 3
**Contribution:** 2
**Rating:** 2
**Confidence:** 4

**Summary:**

The work attempts to provide a accuracy law that relates the relationship between the minimum forecasting error of deep models,  and the complexity of time-series window wise patterns (mainly measured as the variance of the amplitude spectrum distribution). The work is timely and is in a good direction as there have been several papers that develop sophisticated deep time series forecasting models and can only show marginal gains in accuracy on the standard 7-8 datasets such as ETT, electricity, traffic etc.

**Strengths:**

A key use case of the accuracy law as claimed in the work is to test whether saturation wrt accuracy has been achieved for a given dataset, so as to guide the research community towards the need to move to new benchmarks. The work is timely and is in a good direction as there have been several papers that develop sophisticated deep time series forecasting models and can only show marginal gains in accuracy on the standard 7-8 datasets such as ETT, electricity, traffic etc.

**Weaknesses:**

While the work has good motivation, I find the claims to be particularly strong, and not well supported by the theoretical analysis. There are also some concerns regarding the experimental setup.

Despite the claim of finding a novel accuracy law, the actual law proposed in the paper does not seem to have solid theoretical backing or novelty. The work first transforms the time window in to the frequency domain, and then extracts the amplitude spectrum. The variance of this spectrum is deemed to represent the complexity of time window. This measure is then related via a exponential relation to the best MSE achieved for that time series.

This idea is intuitive, but as such not that novel. Several works in time series decompose the data into frequency domain, and it is also straightforward to see the variance of the amplitude spectrum can affect the complexity.

There are some issues that need clarity when testing whether this idea is valid. First, the tests are only done on univariate time series on the LOTSA dataset. Using random sampling over LOTSA dataset, the work claims to create a balanced, diverse and representative dataset to test their law. It is not clear, in a rigorous manner, what “balanced”, “diverse” and “representative” means. How can creating time series on LOTSA dataset be representative of all types of time series found in real world, and how can one be sure that real world time series diversity is represented in the created dataset?  This issue seems like a major limitation which affects the generality of the claim.

Another setting which is hardcoded is the input-96-predict-96 setting. Why this setting was chosen, how does the accuracy law’s performance varies when this time window size varies. One can assume that if the input size increases, then the prior tests for time series complexity, such as ADF, would start working. Currently I could not see this analysis.

Overall, while the claimed law is intuitive, and it is indeed required to establish a metric for complexity of time series, the claims of the work appear overstated, and the accuracy law’s testing is limited by several assumptions made in the experimental setup, which may not hold in wide range of real world settings.

**Questions:**

See the weakness part.

---

> ### Author Response · Authors · 2025-11-29
> **Response to Reviewer cwRd**
>
> We would like to sincerely thank Reviewer cwRd for providing valuable feedback and questions.
>
> > **W1:** "Despite the claim of finding a novel accuracy law, the actual law proposed in the paper does not seem to have solid theoretical backing or novelty. The work first transforms the time window into the frequency domain, and then extracts the amplitude spectrum. The variance of this spectrum is deemed to represent the complexity of the time window. This measure is then related via an exponential relation to the best MSE achieved for that time series."
>
> Thank you for this insightful comment. We would like to highlight that the main novelty of our work lies in **demonstrating a stable exponential relationship between the proposed complexity measure and the achievable forecasting performance** under extensive experiments on a broad set of real‑world datasets using state‑of‑the‑art forecasting models. To our knowledge, such an evaluation has not been documented before.
>
> Also, we want to highlight that even the well-established scaling law for LLMs [1,2], the theoretical foundation is still limited. A robust empirical finding is also valuable in guiding the deep learning practice.
>
> [1] OpenAI, Scaling Laws for Neural Language Models, arXiv 2020
>
> [2] DeepMind, Training Compute-Optimal Large Language Models, arXiv 2022
>
> > **W2:** "This idea is intuitive, but as such not that novel. Several works in time series decompose the data into frequency domain, and it is also straightforward to see the variance of the amplitude spectrum can affect the complexity."
>
> We would like to clarify that **our main contribution does not lie in the frequency decomposition step itself.** Instead, we proposed a novel complexity measurement based on frequency information and revealed an empirical relationship between the proposed complexity and the achievable forecasting performance.
>
> Importantly, our work is not positioned as a theoretical study. Instead, we draw inspiration from the scaling law, which is an empirical investigation designed to uncover generalizable patterns through systematic experimentation. Despite being "intuitive" and focusing on straightforward variables like the number of model parameters and training tokens, the scaling law has had a profound impact on the community. Moreover, we go beyond the empirical findings and further demonstrate how the discovered relationship can guide future research in $\underline{\text{Section 4.2 and Section 4.3 of the original paper}}$.
>
>
> > **W3:** "There are some issues that need clarity when testing whether this idea is valid. First, the tests are only done on univariate time series on the LOTSA dataset. Using random sampling over LOTSA dataset, the work claims to create a balanced, diverse and representative dataset to test their law. It is not clear, in a rigorous manner, what “balanced”, “diverse” and “representative” means. How can creating time series on LOTSA dataset be representative of all types of time series found in real world, and how can one be sure that real world time series diversity is represented in the created dataset?"
>
> Thank you for pointing out the potential issue. LOTSA is a large-scale open time series archive that spans nine major domains, resulting in significant diversity in time series characteristics. However, it is also characterized by an inherent imbalance in both the number of variables and the lengths of the series across different datasets. To avoid this bias, we sample an equal number of series from each dataset so that our experimental subset maintains LOTSA’s cross‑domain diversity while minimizing its imbalance. It is worth noting that no finite benchmark can fully represent all real-world time series.
>
> Nevertheless, our experiments consist of over 2,000 runs and provide broad empirical evidence despite not covering every possible scenario, which aligns with current empirical studies on scaling laws.
>
> > **W4:** "Another setting which is hardcoded is the input-96-predict-96 setting. Why this setting was chosen, how does the accuracy law’s performance varies when this time window size varies. One can assume that if the input size increases, then the prior tests for time series complexity, such as ADF, would start working. Currently I could not see this analysis."
>
>
> Thank you for the insightful questions. The forecasting setup of the input-96-predict-96 setting is a convention of the community. Following your suggestion, we increased the input length to 512 and validated the effectiveness of the proposed complexity against ADF. It has been presented in $\underline{\text{Figure 6 of revised paper}}$.

---

### Official Review · Reviewer_RH7o · 2025-11-02

**Soundness:** 1
**Presentation:** 3
**Contribution:** 2
**Rating:** 2
**Confidence:** 4

**Summary:**

The paper proposes a measure of complexity for time series forecasting tasks. This is verified by showing a log-linear relationship between their proposed measure of complexity and minimum accuracy over 3 deep forecasting models. Specifically, train 3 models on a single univariate time series, and take the minimum (log) MSE/MAE over these 3 models as the dependent variable, and the complexity of that time series as the independent variable, and do a linear regression. Based on this relationship which the paper terms an accuracy law, the paper identifies amongst popular benchmark datasets, which ones are saturated, and which can be further improved on. They also propose a divergence metric inspired by their measure of complexity, and use it to compute the divergence between datasets. Using this, they construct a benchmark with a higher divergence from existing pre-training datasets. Finally, they show that using the measure of complexity to sample more complex time series during pre-training can lead to a model which performs better on this benchmark.

**Strengths:**

The idea of having a measure of complexity which can guide us in developing better benchmarks and training datasets is very promising. The proposed measure of complexity clearly has some ability to be predictive in whether a time series is forecastable or not. The paper also does some interesting work on dataset selection and promisingly shows that when trained on more complex time series, performance improves.

**Weaknesses:**

* The construction of figure 4 and the collection of data points for the proposed law is problematic. The idea is to regress the best possible accuracy on the measure of complexity, however, each data point is only the best metric over 3 models of different architectures. Ideally this should be the best metric over an extensive search of model classes. In fact, the formal statement of the accuracy law claims to be "the minimum forecasting MSE achieved by all feasible deep models", this is a huge overstatement and is not true.
* Along similar lines, there is a huge extrapolation risk between how the datapoints are collected and real world scenarios. The data points are collected in a very artificial setting where 1 deep model is trained on 1 time series, but this is not the case for most real world models, and increasingly so when the paradigm of cross dataset training is becoming more popular. Considerations need to be made about how to measure forecastability, or perhaps we need to be considering conditional forecastability.
* I'm confused as to why the relationship between complexity and performance has been termed an "accuracy law", when it seems that the experiment is simply verifying that the proposed measure of complexity fulfills the desired relationship between complexity and forecastabillity. I presume that the name is inspired by scaling laws, but the term scaling laws comes about from measuring the scale of an aspect of the system and performance, where the way we measure the dependent variable is fixed. However, here, the way the dependent variable is measured is changing based on the independent variable. For example, let's say we want to propose a complexity law, where the independent variable is the measure of complexity of some aspect of the system, e.g. dataset, model arch, and the dependent variable is performance, measured in a fixed manner and not changing based on the complexity measure.
* Given the extrapolation risk, I'm not sure if we can make the conclusion made in 4.2 Practice 1. It also seems to make a presupposition that (deep learning) research is mainly about model design.

Minor
* Some claims made in the introduction are very contentious
  * "For instance, image recognition in computer vision (Deng et al., 2009), a popular and long-standing area since 2000, has a quantifiable and widely accepted goal, that is, to achieve 100% accuracy."  - Even in imagenet, 100% can't be achieved due to labelling errors and inherent ambiguity in images.
  * "Drawing inspiration from other tracks of the machine learning community that never or rarely suffer from such unclear research objectives"  - I'm very surprised by this statement.

**Questions:**

* Could more information about the set up for figure 4 be provided, exactly how the models were trained, what are the model sizes, more information about the time series that were picked for this experiment.
* How are the data points in Figure 6 obtained?
* I'm confused about the divergence metric and figure 7. For Divergence(x, y), are x, y from the same dataset, i.e. TimeBench, or is x from TimeBench and y from Gift-Eval?

---

> ### Author Response · Authors · 2025-11-29
> **Response to Reviewer RH7o (Part 1)**
>
> Many thanks to Reviewer RH7o for providing a detailed and in-depth review, which helped us improve the quality of our submission.
>
> > **W1:** "The idea is to regress the best possible accuracy on the measure of complexity, however, each data point is only the best metric over 3 models of different architectures. Ideally, this should be the best metric over an extensive search of model classes. In fact, the formal statement of the accuracy law claims to be "the minimum forecasting MSE achieved by all feasible deep models", this is a huge overstatement and is not true."
>
> Thank you for the valuable comment. As we stated in $\underline{\text{Line 294 of the origin submission}}$, the three models we selected (DLinear, PatchTST, TimeMixer) are widely recognized as **state‑of‑the‑art architectures** for univariate forecasting and are consistently adopted as competitive baselines in recent works. To further validate that our results are not an artifact of model selection, we conducted an additional experiment incorporating two more recent models (TimeMixerPP and xPatch). Across the 940 datasets, these models outperformed the original three baselines in only 79 and 206 cases, respectively.
>
> Despite these additional wins, the exponential relationship is preserved. The fitted line has a slope of $\alpha = 0.0053$, nearly identical to the $\alpha = 0.0054$  reported in the original submission. This close agreement supports the representativeness of our chosen models and further demonstrates the robustness of the previously observed result.
>
> Also, we want to highlight that, even in the widely acknowledged scaling law, the constant of the fitted curve provided by DeepMind and OpenAI is different. Thus, identifying the exponential relation between MSE and pattern complexity is already a significant advance, which is also well guaranteed after adding new models.
>
> [1] OpenAI, Scaling Laws for Neural Language Models, arXiv 2020
>
> [2] DeepMind, Training Compute-Optimal Large Language Models, arXiv 2022
>
>
> > **W2:** "Along similar lines, there is a huge extrapolation risk between how the datapoints are collected and real world scenarios. The data points are collected in a very artificial setting where 1 deep model is trained on 1 time series, but this is not the case for most real world models, and increasingly so when the paradigm of cross-dataset training is becoming more popular. Considerations need to be made about how to measure forecastability, or perhaps we need to be considering conditional forecastability."
>
> Thank you for the insightful suggestion. We would like to clarify that the experimental setting we use is not an artificial construction, but rather a standard and well‑established paradigm in the field of time series forecasting. Our study is motivated by the empirical observation that, despite substantial architectural innovation, the performance of deep forecasting models on widely used benchmarks has largely stagnated in recent years. To understand whether this stagnation reflects model limitations or fundamental data limitations, our analysis focuses on estimating the intrinsic forecastability of time series data that adhere to the conventional setup. We believe this offers valuable insights into future research.
>
> We fully acknowledge that pre-trained time series foundation models are an emerging trend in real-world applications. However, evaluating forecastability in such settings is currently infeasible because these models are typically trained on undisclosed data, raising concerns about potential data leakage and making it impossible to decouple model performance from overlap with training data.
>
> > **W3:** "I'm confused as to why the relationship between complexity and performance has been termed an "accuracy law", when it seems that the experiment is simply verifying that the proposed measure of complexity fulfills the desired relationship between complexity and forecastability."
>
> Thank you for the instructive comments. We have revised the term "accuracy law" to "complexity law" for clarity. However, we would like to clarify that the dependent variable is the minimum forecasting MSE, which is not changing.

---

> ### Author Response · Authors · 2025-11-29
> **Response to Reviewer RH7o (Part 2)**
>
> > **W4:** "Given the extrapolation risk, I'm not sure if we can make the conclusion made in 4.2 Practice 1. It also seems to make a presupposition that (deep learning) research is mainly about model design."
>
> We would like to highlight that deep time series forecasting models follow a conventional training and evaluation pipeline to ensure a fair comparison [1,2]. Notably, all the experiments in our study strictly follow this well-acknowledged framework. Therefore, our analysis in Practice 1 aims to explain the observation in Figure 1. Specifically, by comparing the optimal performance of deep models on the interested benchmarks against this performance bound estimated by the complexity law, we observe that in certain benchmarks, most of their univariate series fall far below this bound. This empirical, data‑driven observation indicates that the intrinsic predictability limit of the dataset has been reached.
>
> [1] Zhou, Haoyi, et al. "Informer: Beyond efficient transformer for long sequence time-series forecasting." AAAI 2021.
>
> [2] Wu, Haixu, et al. "Timesnet: Temporal 2d-variation modeling for general time series analysis." ICLR 2023.
>
> > **W5:** "Some claims made in the introduction are very contentious“
> >
> > - "For instance, image recognition in computer vision (Deng et al., 2009), a popular and long-standing area since 2000, has a quantifiable and widely accepted goal, that is, to achieve 100% accuracy." - Even in Imagenet, 100% can't be achieved due to labelling errors and inherent ambiguity in images.
> > - "Drawing inspiration from other tracks of the machine learning community that never or rarely suffer from such unclear research objectives" - I'm very surprised by this statement.
>
>  (1) Thank you for pointing this out.  Our intention was not to claim that 100% accuracy on ImageNet is realistically attainable.  What we intended to convey is that fields such as image recognition have a clear and widely understood target performance, whereas such a clear target is rarely available or agreed upon in time series forecasting.
>
> （2）Sorry for the overstatement. Our intended point was that other research areas have clear research objectives for deep models to achieve; time series forecasting lacks a similarly well‑established notion of performance ceilings or difficulty levels for datasets. This absence makes it harder to contextualize model improvements or understand how close current methods are to the intrinsic limits of the data. We have revised the description accordingly.
>
> > **Q1:** "Could more information about the set up for figure 4 be provided, exactly how the models were trained, what are the model sizes, more information about the time series that were picked for this experiment."
>
> Sorry for the missing details. For each series used in the experiments, we split the entire series into train/val/test with the ratio of 7/1/2 following the convention. A total of 940 time series longer than 5000 data points were randomly selected from LoTSA and uniformly truncated to a length of 5000. LoTSA is a large-scale open time series archive that contains time series data across nine domains. The sources of these time series, along with their respective domains and sampling frequencies, are detailed in $\underline{\text{Table 4 of the revised paper}}$.
>
> > **Q2:** "How are the data points in Figure 6 obtained?"
>
> As we stated in $\underline{\text{430 of the main text}}$, the data point in Figure 6 follows the same experiment setup used in Section 3. Each plotted point corresponds to a time series from the benchmark; therefore, the number of points is the number of variates of the benchmark. For each series, we trained advanced models and recorded their minimum forecasting errors, representing the best forecasting performance achieved.
>
> > **Q3:** "I'm confused about the divergence metric and figure 7. For Divergence(x, y), are x, y from the same dataset, i.e. TimeBench, or is x from TimeBench and y from Gift-Eval?"
>
> Sorry for the missing clarification. In Figure 7, x and y come from two different datasets.

---

### Meta-Review · Area_Chair_2HED · 2026-01-07

**Summary:**

This paper addresses an important and timely question: whether performance on standard deep time-series forecasting benchmarks has become saturated. It proposes an empirical relationship between a frequency-based complexity measure and forecasting error, supported by large-scale experiments, and explores implications for benchmark diagnosis and pretraining strategies.

Reviewers generally agree that the motivation is strong, the paper is clearly written, and the empirical effort is substantial. Several reviewers find the discovered correlation promising and potentially useful as a diagnostic signal of forecasting difficulty.

However, a fundamental and recurring concern is that the paper's core claims are significantly overstated relative to the evidence provided. The paper frames the result as a “law” describing the minimum achievable error over all feasible deep models, while in practice, the dependent variable is the minimum performance among a limited, selected set of architectures under a specific protocol. Although the rebuttal adds more models and settings to demonstrate the robustness of the correlation, it does not resolve this conceptual gap. Therefore, the existing experiments cannot fully justify the mentioned claims about intrinsic forecastability, upper bounds, and benchmark saturation.

This paper received 2 scores for rejection, 1 score for marginally below the acceptance threshold, and 2 scores for marginally above the acceptance threshold. The authors' response was extensive and addressed several technical concerns: adding more models (TimeMixerPP, xPatch), testing longer horizons, and providing more experimental details. These actions successfully increased reviewer DZKi's score and solidified the robustness of the observed correlation. However, they did not address the conceptual concerns of Reviewers RH7o and cwRd regarding overstatement, extrapolation, and the very definition of the "law." The central overclaim remains unmitigated. Hence, I recommend rejection at this time, with encouragement for substantial revision and resubmission to a future venue.

**Reviewer Concerns:**

Addressed Main Concerns:
+ Experiments on a longer input horizon.
+ Experiments on a sensitive analysis of the input and output length.
+ The experimental model selection.

However, the main conceptual concern remains outstanding. As pointed by Reviewer cwRd ("How can creating time series on LOTSA dataset be representative of all types of time series found in real world, and how can one be sure that real world time series diversity is represented in the created dataset?"), if the paper continues to frame the observed best-of-selected-models performance as a model-agnostic “law” or intrinsic upper bound on forecastability.

**Reviewer Scores:**

Reviewer VUQD (Score: 6) would like to keep his/her score.

Reviewer DZKi (Score: 6) would like to increase his/her score to 7.

Reviewer B8Rp (Score: 4) would like to keep his/her score.

Reviewer RH7o (Score: 2) would like to keep his/her score.

Reviewer cwRd (Score: 2) would like to keep his/her score.

---

### Decision · Program_Chairs · 2026-01-26

Reject